# Towards Understanding the Universality of Transformers for Next-Token Prediction

**Michaël E. Sander & Gabriel Peyré**
Ecole Normale Supérieure, CNRS
Paris, France
`michael.sander@polytechnique.org, gabriel.peyre@ens.fr`

## Abstract

Causal Transformers are trained to predict the next token for a given context. While it is widely accepted that self-attention is crucial for encoding the causal structure of sequences, the precise underlying mechanism behind this in-context autoregressive learning ability remains unclear. In this paper, we take a step towards understanding this phenomenon by studying the approximation ability of Transformers for next-token prediction. Specifically, we explore the capacity of causal Transformers to predict the next token $x_{t+1}$ given an autoregressive sequence $(x_1, \ldots, x_t)$ as a prompt, where $x_{t+1} = f(x_t)$, and $f$ is a context-dependent function that varies with each sequence. On the theoretical side, we focus on specific instances, namely when $f$ is linear or when $(x_t)_{t \geq 1}$ is periodic. We explicitly construct a Transformer (with linear, exponential, or softmax attention) that learns the mapping $f$ in-context through a causal kernel descent method. The causal kernel descent method we propose provably estimates $x_{t+1}$ based solely on past and current observations $(x_1, \ldots, x_t)$, with connections to the Kaczmarz algorithm in Hilbert spaces. We present experimental results that validate our theoretical findings and suggest their applicability to more general mappings $f$.

## 1 Introduction

The transformative impact of deep learning on artificial intelligence has led to increasingly powerful architectures, with Transformers (Vaswani et al., 2017) at the forefront. These models have in particular revolutionized natural language processing (NLP) (Devlin et al., 2018), and now serve as the foundation for large-scale language models, such as GPT (Radford et al., 2018; Brown et al., 2020), significantly advancing artificial intelligence's capabilities in both understanding and generating human language, setting new benchmarks across various tasks.

Most recent large language models (Hoffmann et al., 2022; Team et al., 2023; Jiang et al., 2023; Dubey et al., 2024) are causal Transformers pretrained to predict the most likely next token $x_{t+1}$ from a finite vocabulary given a context $x_{1:t} = (x_1, \cdots, x_t)$. These models excel at such tasks and beyond, demonstrating what is known as in-context learning: after training, they show remarkable few-shot learning capabilities, inferring patterns from just a few examples within the context (Brown et al., 2020). Recent studies suggest that in-context learning capabilities emerge from the Transformer performing optimization on an inner objective during its forward pass, where the attention matrix plays a crucial role (Von Oswald et al., 2023a; Mahankali et al., 2023; Ahn et al., 2023; Zhang et al., 2023; Kim & Suzuki, 2024). In particular, Cheng et al. (2024) demonstrate that trained Transformers can perform in-context learning through kernel ridge regression. However, the question of why causal Transformers excel at general autoregressive prediction remains open.

In this paper, we propose a kernel interpretation for the autoregressive setting. We introduce a framework to rigorously analyze the expressivity of deep Transformers in next-token prediction. Specifically, we consider sequences generated according to $x_{t+1} = f(x_t)$, with $f$ a context-dependent function in a vector-valued Reproducing Kernel Hilbert Space (RKHS) associated with a positive semi-definite kernel $k$. This generalizes the works of Von Oswald et al. (2023b); Sander et al. (2024). Within this framework, we explore how successive attention layers solve a causal kernel least square regression problem to predict the next token accurately, as described in Figure 1.

Figure 1: Illustration of the method proposed in this paper. Given a sequence $x_{1:t}$, a first layer $\mathcal{T}_0$ computes augmented tokens $e_{1:t}^0$. Next, a stack of $n$ identical Transformer layers $\mathcal{T}$ with residual connections iteratively update the tokens $e_{1:t}^k$, following the causal kernel descent method introduced in Section 4. For autoregressive sequences presented in Assumption 1, and under specific instances outlined in Assumption 2, projecting $e_t^n$ with a projector $P$ yields an estimate $u_t^n$ of $x_{t+1}$ as $n$ and $t$ approach $+\infty$, as stated in Theorem 1.

More precisely, we make the following contributions:

- In Section 3, we formalize the sequence generation model that serves as the foundation for our theoretical results. We then present our main result in Theorem 1, demonstrating that there exists a Transformer model, with an explicit construction, that, given the first $t$ tokens $x_{1:t}$, accurately predicts the next token $x_{t+1}$ as $t$ tends to $+\infty$, for specific instances, namely when $f$ is linear or when the sequence $(x_t)_{t \geq 1}$ is periodic.

- In Section 4, we present the methodology underlying the proof of Theorem 1. We introduce a family of causal kernel descent methods that build an estimate $u_t^\star$ of $x_{t+1}$, based only on past and current observations $x_{1:t}$. This approach modifies a least squares gradient descent method to account for causality while preserving the parallelization benefits of the Transformer architecture. In Theorems 2, 3, and 4, we prove that for the specific cases considered, $u_t^\star - x_{t+1}$ converges to 0 as $t \to +\infty$, drawing connections to the Kaczmarz algorithm (Kaczmarz, 1937) in Hilbert spaces. Finally, Proposition 5 shows that the causal kernel descent methods can be implemented with a Transformer model, as illustrated in Figure 1.

- In Section 5, we first present experimental results that validate our theoretical findings and extend them to a more general class of mappings $f$ beyond those studied in Sections 3 and 4. We then empirically show that the Transformer models constructed in Theorem 1 can be successfully fine-tuned to obtain faster convergence of the estimate with the sequence length $t$.

## 2 BACKGROUND AND RELATED WORK

**Transformers.** Transformers (Vaswani et al., 2017) process sequences of tokens $(x_1, \cdots, x_T)$ or arbitrarily length $T$. After embedding the sequence into a new sequence $(e_1, \cdots, e_T)$, the Transformer consists of a series of blocks with residual connections (He et al., 2016). Each block comprises two primary components: a multi-head self-attention mechanism and a feedforward multi-layer perceptron, with the latter operating independently on each token. We will almost always disregard the feedforward layer in this study (except when building augmented tokens), as is common in theoretical analyses of in-context learning (Mahankali et al., 2023; Ahn et al., 2023; Zhang et al., 2023). In contrast, the multi-head self-attention mechanism involves pairwise interaction between the tokens. This module consists in applying multiple self-attention operations in parallel, parametrized by a set of weight matrices $(W_Q^h, W_K^h, W_V^h)_{1 \leq h \leq H}$, where $H$ denotes the number of attention heads (Vaswani et al., 2017; Michel et al., 2019). The output of the multi-head self-attention mechanism is given by:

$$e_t \leftarrow e_t + \mathcal{T}(e_{1:t}), \quad \text{with} \quad \mathcal{T}(e_{1:t}) := \sum_{h=1}^{H} \sum_{s=1}^{t} \mathcal{A}_{t,s}^h W_V^h e_s, \tag{1}$$

where $\mathcal{A}^h$, the attention matrix, determines the attention weights between tokens and is typically defined as: $\mathcal{A}_{t,:}^h = \mathcal{N}(\langle W_Q^h e_t, W_K^h e_: \rangle)$, with $\langle \cdot, \cdot \rangle$ representing a dot product and $\mathcal{N}$ being a normalization function. The standard choice is to consider $\mathcal{N} = \text{softmax}$, i.e.

$$\mathcal{A}_{t,s}^h = e^{\langle W_Q^h e_t, W_K^h e_s \rangle} / \sum_{\tau=1}^{t} e^{\langle W_Q^h e_t, W_K^h e_\tau \rangle}.$$

One can also consider the unnormalized attention when $\mathcal{N} = \exp$. Another approach is to consider $\mathcal{N} = \text{id}$, which corresponds to what is known (despite being non-linear) as linear attention

(Katharopoulos et al., 2020), enabling faster inference. One significant advantage of equation 1 is that the updates on the $e_t$'s can be computed in parallel during training, leveraging modern hardware for faster computations.

**Expressivity.** The universal approximation properties of encoder-only Transformers are well established. Yun et al. (2019); Nath et al. (2024); Furuya et al. (2024) demonstrate that Transformers can approximate permutation-equivariant functions. A more constructive approach, though applicable to a narrower class of functions, is proposed by Wang & E (2024). When it comes to decoder-only models, the expressivity of Transformers for next-token prediction is not yet fully understood, though a popular recent line of works studies the in-context learning ability of Transformers.

**In-context learning.** A major property of Transformers is that they adapt their computations given the context. In particular, given a context $(x_1, g(x_1), \cdots, x_n)$, a trained large Transformer can infer the next output $g(x_n)$ without parameter updates. Many recent studies have contributed to understanding this phenomenon. The seminal work of Von Oswald et al. (2023a) considers functions $g$ of the form $g(x) = w^\top x$ for some $w$ and construct a linear Transformer for which the forward pass is equivalent to a single step of gradient descent on a mean squared error loss. Theoretical guarantees are provided by Mahankali et al. (2023); Ahn et al. (2023); Zhang et al. (2023), showing a trained one-layer linear Transformer implements one step of (preconditioned) gradient descent. Other works study the softmax attention without accounting for training dynamics (Garg et al., 2022; Akyürek et al., 2022; Li et al., 2023). Of particular interest to us is the recent work of Cheng et al. (2024) which shows that there exists a simple parameter configuration of non-linear Transformers such that they implement gradient descent in the function space with respect to the RKHS metric induced by the attention kernel. In this work, we take a step further by considering a more general next-token prediction task and propose a causal kernel descent method that can be implemented by a Transformer to solve it. For this, we extend the autoregressive in-context learning setting introduced by Sander et al. (2024), where tokens are generated according to an autoregressive process of order 1: $x_{t+1} = f(x_t)$, where $f(x) = Wx$ for a context-dependant parameter $W$, varying with each sequence. The autoregressive in-context learning ability is described as the model's capacity to decompose its prediction into two steps: first, estimating $W$ through an in-context mapping, and then applying a straightforward prediction function, which is either equal to or closely related to $x \mapsto Wx$. However, Sander et al. (2024) focus solely on linear Transformers, whereas our study encapsulates linear, exponential, and softmax Transformers. Our key contribution is showing how such Transformers can implement an optimization algorithm–termed causal kernel descent–within their forward pass. This can be interpreted as a mesa-optimization mechanism (Von Oswald et al., 2023b).

**Neural Ordinary Differential Equations.** Neural ODEs (Weinan, 2017; Chen et al., 2018) are a class of implicit models where a neural network $\mathcal{F}$ parameterizes a vector field in an ordinary differential equation (ODE), as follows:

$$\frac{de}{d\tau}(\tau) = \mathcal{F}(e(\tau)), \tag{2}$$

where $\tau$ denotes the continuous depth of the network. For a given input $e(0)$, a neural ODE outputs a deep representation $e^\star$, which is the solution (when it exists) of equation 2 with initial condition $e(0)$, at some finite or infinite time horizon. Neural ODEs can be viewed as a continuous-depth analog of deep Residual Networks (He et al., 2016), where the latter corresponds to an Euler discretization for solving equation 2 (Marion et al., 2023). As such, neural ODEs are widely employed to better understand the theoretical properties of Residual Networks. In the context of Transformers, the neural ODE framework has emerged as a valuable tool for studying attention-based models, such as the impact of the choice of normalization function $\mathcal{N}$ on model behavior (Sander et al., 2022) and the emergence of clusters (Geshkovski et al., 2024). However, these studies do not address the autoregressive setting, which we tackle in this work.

## 3 TRANSFORMERS FOR AUTOREGRESSIVE IN-CONTEXT LEARNING

**Notations.** Throughout the paper, we denote $d$ the dimension of $x_t$, $\|.\|$ the $\ell_2$ norm, $O(d)$ the orthogonal manifold, $S^{d-1}$ the unit sphere in $\mathbb{R}^d$ and $^*$ the adjoint.

In this section, we describe the types of autoregressive sequences we consider and present our main results, which show that we can explicitly construct Transformer models that approximate the next token in the sequence.

### 3.1 PROPOSED FRAMEWORK

**Reproducing Kernel Hilbert Space.** For $k : \mathbb{R}^d \times \mathbb{R}^d \to \mathbb{R}$ a positive definite kernel, we define $\mathcal{H}$ as the vector-valued Reproducing Kernel Hilbert Space (RKHS) associated with the feature map $\varphi : \mathbb{R}^d \to \mathcal{H}$, where $\varphi(x) = k(x, \cdot)$. Therefore, for any function $f \in \mathcal{H}$, there exists a linear map $W : \mathcal{H} \to \mathbb{R}^d$ such that $f(x) = W\varphi(x)$, where, denoting $W = (w_1, \cdots, w_d)$ as $d$ vectors of $\mathcal{H}$, $\|W\|_{\mathcal{H}} := \sum_{i=1}^{d} \|w_i\|_{\mathcal{H}}^2$ is finite.

We consider autoregressive sequences of order $1$, which we formalize in the following assumption.

**Assumption 1** (Autoregressive sequences.)**.** *We consider sequences of order* $1$*, defined as follows:*

- ***Initial State****: The sequence starts with some* $x_1 \in S^{d-1}$*.*

- ***Hidden Variable****: We suppose there is a hidden variable* $f \in \mathcal{H}$ *such that the subsequent states are generated autoregressively as* $x_{t+1} = f(x_t)$ *for* $t \geq 1$*.*

Note that although we focus on first-order recursions here, higher-order recursions can be considered by embedding tokens in a higher-dimensional space. Indeed, for recursions of the form $x_{t+1} = g(x_t, \cdots, x_{t-\tau})$ for some context mapping $g$, defining $y_t := (x_t, \cdots, x_{t-\tau}) \in \mathbb{R}^{(\tau+1)d}$, one has $y_{t+1} = (x_{t+1}, \cdots, x_{t+1-\tau}) = (g(x_t, \cdots, x_{t-\tau}), \cdots, x_{t+1-\tau})$ which only depends on $y_t$. We thus have $y_{t+1} = f(y_t)$ for some mapping $f$. Therefore as long as the recursion memory is finite, our approach can be generalized. The formulation in Assumption 2 differs from the classical in-context learning setup, where sequences consist of input-output pairs $(x_i, y_i)$, that are often supposed to be iid. We argue that modeling sequences with $x_{t+1} = f(x_t)$ better reflects the nature of real-world sequences on which causal Transformers, such as large language models (LLMs), are trained. Indeed, our proposed model incorporates autoregressive relationships, aligning more closely with the data LLMs encounter during training. However, similarly to in-context learning, in order to accurately predict the next token $x_{t+1}$ given the previous states $x_{1:t}$ as inputs, a Transformer would have to implicitly estimate the hidden map $f$. In this paper, we propose a general method to provide an estimate of $x_{t+1}$ given $x_{1:t}$. However, to prove that such an estimate can be built with a standard Transformer, and in order to prove our universality results in Theorem 1, we consider specific choices for the kernel $k$ and the function $f$, summarized in the following assumption.

**Assumption 2** (Specific considerations.)**.** *We define the following instances.*

*(1)* $k(x, y) = k_{id}(x, y) := \langle x, y \rangle$, $f(x) = Wx$ *for some* $W \in O(d)$*.*

*(2)* $k(x, y) = k_{exp}(x, y) := e^{\langle x, y \rangle}$ *and* $f(x) = \Omega x$ *for some* $\Omega \in O(d)$*.*

*(3)* $k = k_{exp}$, *the sequence* $(x_t)_{t \geq 1}$ *is periodic and* $x_t \in S^{d-1}$ *for all* $t$*.*

Note that for these specific kernel choices, because $\|x_t\| = 1$, one has $k(x_t, x_t) = k(x_1, x_1)$ for all $t$. This follows because $k(x_t, x_t)$ depends only on $\|x_t\| = \|x_1\|$.

**Augmented tokens.** A crucial step in our construction is the building of augmented tokens for a sequence $(x_1, \cdots, x_T)$. Indeed, in Von Oswald et al. (2023b) and Sander et al. (2024), augmented tokens are used, explicitly encoding the relative positions of tokens $x_t$. This approach is similar to in-context learning, where sequences often consist of pairs $(x_i, y_i)$ to encode both input and output information. However, such a construction with attention-based models is non-trivial. Sander et al. (2024) propose a method for constructing augmented tokens using general positional encoding. In this work, we provide a more detailed result, showing that augmented tokens can be computed with a one-layer Transformer that employs dot-product absolute positional encoding. We introduce a beginning-of-sequence token $x_0 := 0_d$ and consider the extended sequence $x_{0:T} = (x_0, x_1, \cdots, x_T)$. In this work, we are going to consider the augmented tokens $e_t^0 \in \mathbb{R}^{4d+2}$ defined as

$$e_t^0 := (x_{t-1}, 0, x_t, 1, x_t, 0_d) \text{ for } t > 1, \text{ and } e_1^0 := (0_d, 1, x_t, 1, 0_d, 0_d).$$

The intuition behind augmented tokens is that they enable the model to directly access two successive elements in the sequence. Importantly, while prior works, such as Von Oswald et al. (2023a;b), directly worked on augmented tokens, one of our contribution lies in showing how they can be computed using positional encoding-based attention. The augmented tokens incorporate standard

beginning-of-sequence tokens, while the 1's correspond to concatenated positional encodings, which are technically required in our proofs. More precisely, the following proposition shows that the $e_t^0$ can be approximated with a Transformer layer.

**Proposition 1.** *There exists a sequence of one-layer and 2-heads causal Transformer $\mathcal{T}_0^n$ with $\mathcal{N} = softmax$ followed by a feedforward layer, such that $\mathcal{T}_0(x_{0:t}) := \lim_{n \to +\infty} \mathcal{T}_0^n(x_{0:t}) = e_t^0$.*

A proof of this result, along with the explicit construction of the corresponding model, is provided in Appendix A.1. Here, $e_1^0$ should be interpreted as a new beginning-of-sequence token. The 0's and 1's in $e_t^0$ correspond to specific choices of positional encodings, which are crucial for handling the softmax normalization in our main Theorem 1. From this point onward, we will consider the new augmented tokens $e_t^0$.

## 3.2 MAIN RESULT

We now present our main theorem. Using the augmented tokens $e_t^0$, we demonstrate that a stack of Transformer layers can approximate the next token $x_{t+1}$ based solely on $e_{1:t}^0$ (and thus solely on $x_{1:t}$ as well) as $t$ increases.

**Approximation with Transformers.** We consider a model $\mathcal{M}^n$ composed of $n$ identical Transformer layers $\mathcal{T}$ with residual connections, followed by a projection. The model iterates from the beginning-of-sequence token $e_t^0$ as:

$$\mathcal{M}^n(x_{1:t}) := Pe_t^n, \quad \text{where} \quad e_t^{k+1} = e_t^k + \mathcal{T}(e_{1:t}^k), \quad 0 \le k \le n-1, \tag{3}$$

with $P : \mathbb{R}^{4d+2} \to \mathbb{R}^d$ the projector selecting the last $d$ coordinates, which can be integrated in the form of a token-wise feedforward layer.

We have the following main theorem, which shows that for the specific instances we consider, there exist Transformer layers $\mathcal{T}$ such that $\lim_{t \to +\infty} \lim_{n \to +\infty} \mathcal{M}^n(x_{1:t}) - x_{t+1} = 0$.

**Theorem 1** (On the expressivity of Transformers for Next Token Prediction). *For each instance in Assumption 2, there exists an attention-only, one-layer, two-head causal Transformer $\mathcal{T}$ with attention normalization $\mathcal{N}$ such that, for any autoregressive sequence $(x_t)_{t \ge 1}$ generated according to Assumption 1, $\mathcal{M}^n(x_{1:t})$ converges exponentially fast as $n$ goes to infinity. Furthermore, denoting $\mathcal{M}(x_{1:t}) := \lim_{n \to +\infty} \mathcal{M}^n(x_{1:t})$, one has $\lim_{t \to +\infty}(\mathcal{M}(x_{1:t}) - x_{t+1}) = 0$. More specifically:*

- *For instance (1), $\mathcal{N} = id$, and the convergence of $(\mathcal{M}(x_{1:t}) - x_{t+1})$ to 0 is exponentially fast in $t$ for almost all $x_1$ and $W$.*

- *For instance (2), $\mathcal{N} = exp$ or $\mathcal{N} = softmax$.*

- *For instance (3), $\mathcal{N} = exp$ or $\mathcal{N} = softmax$, and the convergence of $(\mathcal{M}(x_{1:t}) - x_{t+1})$ to 0 is exponentially fast in $t$.*

*Finally, for any of the instances above, when $\mathcal{N} = id$ or $\mathcal{N} = exp$, we have $\mathcal{M}^n(x_{1:t}) = \mathcal{M}(x_{1:t})$ as long as $n \ge t$, so that $\lim_{n,t \to +\infty, \, n \ge t}(\mathcal{M}^n(x_{1:t}) - x_{t+1}) = 0$.*

We emphasize that the models $\mathcal{T}$ correspond to explicit constructions. Specifically, we have:

$$W_Q^1, W_K^1 \in \mathbb{R}^{d \times (4d+2)}, \quad W_V^1, W_V^2 \in \mathbb{R}^{(4d+2) \times (4d+2)}, \quad W_Q^2, W_K^2 \in \mathbb{R}^{(d+1) \times (4d+2)}.$$

More precisely, one has

$$W_Q^{(1)} = W_K^{(1)} = [\mathbf{0}_{d \times d+1}, I_d, \mathbf{0}_{d \times 2d+1}],$$

$$W_V^{(1)} = - \begin{pmatrix} \mathbf{0}_{d+1 \times d+1} & \mathbf{0}_{d+1 \times d+1} & \mathbf{0}_{d+1 \times d} & \mathbf{0}_{d+1 \times d} \\ \mathbf{0}_{d+1 \times d+1} & \mathbf{0}_{d+1 \times d+1} & \mathbf{0}_{d+1 \times d} & \mathbf{0}_{d+1 \times d} \\ \mathbf{0}_{d \times d+1} & \mathbf{0}_{d \times d+1} & \mathbf{0}_{d \times d} & \mathbf{0}_{d \times d} \\ \mathbf{0}_{d \times d+1} & \mathbf{0}_{d \times d+1} & \mathbf{0}_{d \times d} & I_d \end{pmatrix},$$

$$W_Q^{(2)} = [\mathbf{0}_{d+1 \times d+1}, I_{d+1}, \mathbf{0}_{d+1 \times 2d}],$$

$$W_K^{(2)} = [I_{d+1}, \mathbf{0}_{d+1 \times d+1}, \mathbf{0}_{d+1 \times 2d}],$$

$$W_V^{(2)} = \begin{pmatrix} \mathbf{0}_{d+1 \times d+1} & \mathbf{0}_{d+1 \times d+1} & \mathbf{0}_{d+1 \times d} & \mathbf{0}_{d+1 \times d} \\ \mathbf{0}_{d+1 \times d+1} & \mathbf{0}_{d+1 \times d+1} & \mathbf{0}_{d+1 \times d} & \mathbf{0}_{d+1 \times d} \\ \mathbf{0}_{d \times d+1} & \mathbf{0}_{d \times d+1} & \mathbf{0}_{d \times d} & \mathbf{0}_{d \times d} \\ \mathbf{0}_{d \times d+1} & \mathbf{0}_{d \times d+1} & I_d & \mathbf{0}_{d \times d} \end{pmatrix}.$$

The proof of Theorem 1, which is in Appendix A.9, relies on demonstrating that the model $\mathcal{M}^n$ implements a causal kernel descent method, which we describe and analyze in Section 4.

**Remark 1.** *In Theorem 1, we have $\|\mathcal{M}^n(x_{1:t}) - x_{t+1}\| \leq \varepsilon_1(n,t) + \varepsilon_2(t)$, with $\varepsilon_1(n,t) = \|\mathcal{M}^n(x_{1:t}) - \mathcal{M}(x_{1:t})\|$ and $\varepsilon_2(t) = \|\mathcal{M}(x_{1:t}) - x_{t+1}\|$. The proof of Theorem 1 reveals that $\lim_{n\to+\infty} \varepsilon_1(n,t) = 0$ and $\lim_{t\to+\infty} \varepsilon_2(t) = 0$. When $\mathcal{N} = id$ or $\mathcal{N} = exp$, $\varepsilon_1(n,t) = 0$ if $n \geq t$, hence the last statement of the theorem. When $\mathcal{N} = softmax$, we conjecture that $\varepsilon_1(n,t) \to 0$ as $n, t \to +\infty$ and $t/n \to 0$, which implies $\lim_{n,t\to+\infty \text{ and } t/n\to0}(\mathcal{M}^n(x_{1:t}) - x_{t+1}) = 0$. We provide evidence for this conjecture in the last paragraph of Appendix A.9.*

The model $\mathcal{M}$ introduced in Theorem 1 also has a continuous-time interpretation, which we now formulate.

**Neural ODE Interpretation.** The model $\mathcal{M}$ defined in Theorem 1 as the limit when $n$—the number of Transformer layers—goes to infinity can be interpreted as a continuous-time neural ODE (Chen et al., 2018). Specifically, $\mathcal{M}$ satisfies

$$\mathcal{M}(x_{1:t}) := \lim_{\tau\to+\infty} Pe_t(\tau), \quad \text{where} \quad \frac{de_t}{d\tau}(\tau) = \mathcal{T}(e_{1:t}(\tau)) \quad \text{with} \quad e_t(0) = \mathcal{T}_0(x_{0:t}). \quad (4)$$

We now present the theory behind our constructions to guarantee consistent approximation of the next token $x_{t+1}$ as the sequence size increases.

**Comparison with RNNs.** Even though we consider autoregressive sequences, it is not straightforward that recurrent neural networks (RNNs) can effectively capture these models. This is because estimating $W$ in-context requires computations with long-range dependencies. Determining the optimal $W$ indeed requires inverting the data covariance matrix. Attention mechanisms inherently handle such procedure, which is proven in this work. We believe RNNs would require more layers to "propagate" such information. While each RNN layer is less computationally expensive than an attention layer, the overall cost might be similar. Investigating this is complex and beyond the scope of this article, which demonstrates how current Transformer-based architectures are particularly well-suited for in-context learning due to their global attention mechanism.

## 4 CAUSAL KERNEL DESCENT

In this section, we introduce a causal method to instantiate the iterations in equation 3 and prove Theorem 1. Specifically, we propose a causal kernel descent method that incorporates causality into standard gradient descent for least squares minimization.

### 4.1 CAUSAL DESCENT

**Non-Causal Descent.** We consider a sequence $x_{1:T} := (x_1, \cdots, x_T) \in \mathbb{R}^{T \times d}$. The goal is to solve the least squares minimization problem of minimizing $\sum_{s=1}^{T-1} \|f(x_s) - x_{s+1}\|^2$ with respect to $f \in \mathcal{H}$. Recall that for any $f \in \mathcal{H}$, there exists a linear map $W : \mathcal{H} \to \mathbb{R}^d$ such that $f(x) = W\varphi(x)$ for all $x \in \mathbb{R}^d$. Thus, we consider the least squares optimization problem:

$$\min_W E(W) := \sum_{s=1}^{T-1} \|W\varphi(x_s) - x_{s+1}\|^2. \quad (5)$$

We solve equation 5 using gradient descent with step size $\frac{\eta}{2}$, starting from an initial $W^0$ and iterating for $0 \leq k \leq n-1$:

$$W^{k+1} = W^k - \eta \sum_{s=1}^{T-1} (W^k\varphi(x_s) - x_{s+1})\varphi(x_s)^*.$$

We define a prediction variable $u_s^k := W^k\varphi(x_s)$. By right-multiplying the gradient descent equation by $\varphi(x_t)$, we obtain:

$$u_t^{k+1} = u_t^k - \eta \sum_{s=1}^{T-1} k(x_t, x_s)(u_s^k - x_{s+1}), \quad (6)$$

which corresponds to a least squares descent on the predictions. However, this descent is non-causal because the update of $u_t$ depends on $u_{1:T-1}$ and $x_{1:T}$, making it unsuitable for implementation in a causal Transformer. We now propose a causal formulation for equation 6.

**Causal Descent.** Inspired by the descent in equation 6, we propose a modified least squares descent that introduces causality, ensuring that each estimate of $x_{t+1}$ is based solely on past and current observations $x_{1:t}$. To achieve this, we define the following *causal kernel descent*, which incorporates both an unnormalized and a row-wise normalized framework. Starting from any initial $u_t^0$, the descent iterates for $0 \le k \le n-1$ as follows:

$$u_t^{k+1} = u_t^k - \eta \sum_{s=1}^{t} A_{t,s}(u_s^k - 1_{s<t}x_{s+1}) \text{ with } A_{t,s} = \begin{cases} k(x_t, x_s), & \text{if } k = k_{\text{id}} \\ k(x_t, x_s) \text{ or } \frac{k(x_t,x_s)}{\sum_{\tau=1}^{t} k(x_t,x_\tau)}, & \text{if } k = k_{\text{exp}} \end{cases}.$$
(7)

We denote $A$ the corresponding lower triangular matrix. Note that this descent is causal in the sense that $u_t^n$ depends only on $x_{1:t}$. Note also that each iteration in equation 7 can be parallelized.

For well-chosen step sizes $\eta$, the method in equation 7 converges. Specifically, we have the following proposition:

**Proposition 2.** *For each causal kernel descent in equation 7, there exists $\eta^\star$ and $u_t^\star$ such that, for all $0 < \eta < \eta^\star$, $u_t^n \to u_t^\star$ exponentially fast as $n$ goes to infinity. Specifically:*

- *When $A_{t,s} = k(x_t, x_s)$, then $\eta^\star = \frac{2}{k(x_1,x_1)}$. Moreover, when $\eta = \frac{1}{k(x_1,x_1)}$, $u_t^n = u_t^\star$ if $n \ge t$.*

- *When $A_{t,s} = \frac{k(x_t,x_s)}{\sum_{\tau=1}^{t} k(x_t,x_\tau)}$, then $\eta^\star = 2$.*

Our proof, in Appendix A.2, relies on the lower triangular property of the matrix $A$.

Our objective is to show that $u_t^\star$ is "close" to $x_{t+1}$ when $t$ is sufficiently big, demonstrating that the causal kernel descent accurately tracks the future states provided it has seen sufficiently long context. We have the following result.

**Proposition 3.** *Let $(\mu_t)_{t \ge 1}$ be the unique sequence of vectors satisfying*

$$\sum_{s=1}^{t} \mu_s k(x_s, x_t) = x_{t+1}, \quad \forall t \ge 1.$$
(8)

*Then for all $t \ge 1$, $x_{t+1} - u_t^\star = k(x_1, x_1)\mu_t$.*

For a proof, see Appendix A.3. From Proposition 3, we see that $\lim_{t \to +\infty}(u_t^\star - x_{t+1}) = 0$ is equivalent to having $\lim_{t \to +\infty} \mu_t = 0$. We provide a dual interpretation for $\mu$ in Appendix B.

### 4.2 CONVERGENCE IN THE SEQUENCE LENGTH

In this section, we prove that under the instances defined in Assumption 2, $\lim_{t \to +\infty}(u_t^\star - x_{t+1}) = 0$. For any $\mu$ satisfying equation 8, we define the following map from $\mathcal{H}$ to $\mathbb{R}^d$:

$$W_t := \sum_{s=1}^{t} \mu_s \varphi(x_s)^*.$$
(9)

By construction, we have for all $t$: $W_t \varphi(x_t) = x_{t+1}$. To prove that $\lim_{t \to +\infty} \mu_t = 0$ (and therefore $\lim_{t \to +\infty}(u_t^\star - x_{t+1}) = 0$), we will actually prove a stronger result, showing that for the instances in Assumption 2 and in a specific limit sense, $W_t \to W$. For this, we define

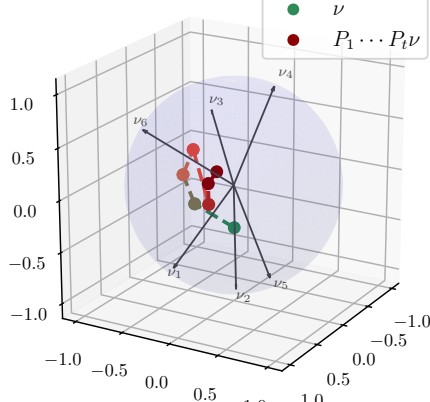

$$\nu_t := \frac{\varphi(x_t)}{\sqrt{k(x_t, x_t)}} = \frac{\varphi(x_t)}{\|\varphi(x_t)\|_{\mathcal{H}}} \quad \text{and} \quad P_t := (I - \nu_t \nu_t^*) \cdot$$
(10)

Figure 2: For some random vectors $\nu$ (green) and $\nu_1, \cdots \nu_6$ in $S^2$, we display $P_6 \nu$ (grey), $P_5 P_6 \nu$ (orange), ... and $P_1 \cdots P_6 \nu$.

Therefore, $P_t$ is the orthogonal projection onto the subspace orthogonal to $\varphi(x_t)$. With these notations, we have the following proposition:

**Proposition 4** (Estimate Update Recursion). *For all $t \geq 1$, we have*

$$W_t - W = -W P_1 P_2 \cdots P_t.$$

For a proof, see Appendix A.4. Intuitively, the recursive relationship in Proposition 4 shows how the difference between $W_t$ and $W$ is progressively reduced by successive orthogonal projections. Such a process is illustrated in Figure 2, where each point corresponds to an iterate $\nu_t$, defined as $\nu_t = P_1 \cdots P_t \nu$. Note that without further assumptions, there is no guarantee that $(W_t - W)\varphi(x)$ converges to $0$ for arbitrary $x$. However, for the specific instances considered in Assumption 2, we are able to establish convergence for vectors $x$ in a certain space.

We first focus on instance (1), where we can derive convergence speed results.

**Linear recursions and dot-product kernel.** In this section, we consider instance (1) from Assumption 2. Under these assumptions, we prove the following theorem:

**Theorem 2** ($k = k_{\mathrm{id}}$, linear recursions). *Under instance* (1) *in Assumption 2, one has that* $\lim_{t \to +\infty}(u_t^\star - x_{t+1}) = 0$. *In addition, for almost all $W$ and $x_1$, $W_t \to W$ exponentially fast.*

*Proof sketch.* We first show that $W_t - W = -(W(I - x_1 x_1^\top))^t W^{-t+1}$. We then establish that for almost all $W$ and $x_1$, $\rho(W(I - x_1 x_1^\top)) < 1$, where $\rho$ denotes the spectral radius. □

See Appendix A.5 for a full proof. We now turn to the case where $k = k_{\exp}$.

**Linear recursions and exponential kernel.** In this paragraph, we consider instance (2) in Assumption 2. We present the following theorem, with a complete proof provided in Appendix A.6.

**Theorem 3** ($k = k_{\exp}$, linear recursions). *Under instance* (2) *in Assumption 2, for any $x \in \mathbb{R}^d$, $(W_t^* - W^*)x \to 0$. In particular, $\lim_{t \to +\infty}(u_t^\star - x_{t+1}) = 0$.*

*Proof sketch.* We consider the subset of $\mathcal{H}$ comprising functions $x \mapsto \sum_{s=1}^{\tau} a_s \nu_s$, for $a_s \in \mathbb{R}^d$ and $\tau \geq 1$. This subset is a pre-Hilbert space under the inner product inherited from $\mathcal{H}$. By completing this space with respect to the induced norm from $\mathcal{H}$, we obtain a new Hilbert space $\mathcal{H}'$. We show that $P_t P_{t-1} \cdots P_1 \to 0$ strongly in $\mathcal{H}'$ as $t \to +\infty$. For this, we observe that the convergence of $P_t P_{t-1} \cdots P_1$ is equivalent to the convergence of the Kaczmarz algorithm (Kaczmarz, 1937). A sequence $(\nu_s)_{s \geq 1}$ of unit vector for which $P_t P_{t-1} \cdots P_1 \to 0$ strongly as $t \to +\infty$ is referred to as *effective*. The specificities of our case are twofold: first, $\nu_s \in \mathcal{H}'$, which is potentially of infinite dimension, and second, the vectors $\nu_s$ follow an autoregressive relation. We show that the sequence $(\nu_s)_{s \geq 1}$ is effective in $\mathcal{H}'$. Note that because $\Omega \in O(d)$, one has for any positive integers $t, s, r$ that $\langle \nu_{s+r}, \nu_{t+r} \rangle = \langle \nu_s, \nu_t \rangle$. Such sequence is called *stationary*. Bochner's theorem states that there exists a measure $\sigma$ on the unit circle $S^1$–called *spectral measure*–such that, for all $t \geq 1$,

$$a_t := \langle \nu_{t+1}, \nu_1 \rangle = \int_{S^1} z^t \, d\sigma(z).$$

We then use the following characterization from Kwapień & Mycielski (2001); Rainis Haller (2005).

**Theorem** (Effectiveness of stationary sequences (Kwapień & Mycielski, 2001)). *A stationary sequence of unit vectors which is linearly dense in a Hilbert space is effective if and only if its spectral measure either coincides with the normalized Lebesgue measure or is singular with respect to the Lebesgue measure.*

This characterization combined with Fourier analysis results shows that the sequence $(\nu_s)_{s \geq 1}$ is effective. While this does not necessarily imply the strong convergence of $W_t - W = -W P_1 \cdots P_t$ to $0$, this is enough to prove that $\lim_{t \to +\infty}(u_t^\star - x_{t+1}) = 0$. □

**Periodic recursions.** In this paragraph, we turn to instance (3) in Assumption 2, where $k = k_{\exp}$ and the sequence $(x_t)_{t \geq 1}$ is assumed to be periodic. The motivation for considering periodic sequences is to better align with real-world next-token prediction tasks, where the vocabulary has a finite size and is frequently repeated in cyclical patterns. We have the following theorem (proof in Appendix A.7).

**Theorem 4** ($k = k_{\exp}$, periodic recursions.). *Under instance* (3) *in Assumption 2, one has that* $\lim_{t \to +\infty}(u_t^\star - x_{t+1}) = 0$ *exponentially fast.*

### 4.3 IMPLEMENTATION WITH TRANSFORMERS

We can now use the previous results stating that the output of the causal kernel method $u_t^\star$ approaches $x_{t+1}$ as $t$ increases to construct the models presented in Section 3.

**Expressing equation 7 with a Transformer.** We have the following proposition, which, combined with Theorems 2, 3 and 4, allow us to prove Theorem 1.

**Proposition 5.** *For any $\eta > 0$, for each configuration of equation 7, there exists an attention-only, one-layer, two-head causal Transformer $\mathcal{T}$ with attention normalization $\mathcal{N}$ such that, for any autoregressive sequence $(x_t)_{t\geq 1}$ generated according to Assumption 1, defining $e_t^k := (x_{t-1}, 0, x_t, 1, x_t, u_t^k)$ for $t > 1$ and $e_1^k := (0_d, 1, x_t, 1, 0_d, u_t^k)$, $e_{1:t}^{1:n}$ solves equation 3 if and only if $u_{1:t}^{1:n}$ solves equation 7. More specifically,*

- *When $A_{t,s} = k(x_t, x_s)$ and $k = k_{id}$, we have $\mathcal{N} = id$.*

- *When $A_{t,s} = k(x_t, x_s)$ and $k = k_{exp}$, we have $\mathcal{N} = exp$.*

- *When $A_{t,s} = \frac{k(x_t,x_s)}{\sum_{\tau=1}^{t} k(x_t,x_\tau)}$ and $k = k_{exp}$, we have $\mathcal{N} = softmax$.*

Here again, we stress that the transformer layers $\mathcal{T}$ correspond to an explicit construction. See Appendix A.8 for a constructive proof. With Proposition 5, we prove Theorem 1, in Appendix A.9.

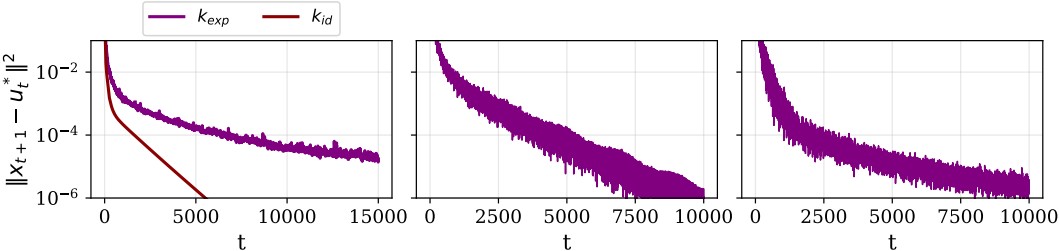

Figure 3: **Evolution** of the squared error $\|u_t^\star - x_{t+1}\|^2$ with $t$ for different scenarios. The curves are averaged over five sequences $x_{1:t}$. Left: instances (1) and (2) (with random $W$ and $\Omega$), illustrating Theorems 2 and 3 ($d = 15$). Center: instance (3), illustrating Theorem 4 ($d = 15$, the period $t_p$ is randomly sampled between 20 and 40, and a random sequence is repeated $t_p$ times). Right: instance (4) described in Section 5 ($d = 4$).

## 5 EXPERIMENTS

In this section, we present experimental results. Our code will be open-sourced.

**Illustration of the Theorems.** We first illustrate the theoretical results of Theorems 2, 3, and 4. For the three instances in Assumption 2, we compute $u_t^\star$ as defined in Proposition 2. We then examine the evolution of $\|u_t^\star - x_{t+1}\|^2$ with $t$. The corresponding curves are shown in Figure 3 (left and center), illustrating the convergence of $u_t^\star - x_{t+1}$ to zero for the considered instances. As predicted by our theory, the convergence is exponential for instances (1) and (3).

**More Complex Iterations.** We also explore the potential convergence of $u_t^\star - x_{t+1}$ to zero for a new instance (4), described as follows. The key idea is to apply "non-linear rotations" to 2-dimensional chunks of the input. Each of these rotations is parameterized by an angle $\theta$ and a scalar $q$, and defined as $z \mapsto R_\theta z^q$, where $R_\theta$ is a 2 dimensional rotation of angle $\theta$. We now formalize this idea. We consider sequences $\{x_t\}$ in $\mathbb{R}^{2p}$ generated according to the process in Assumption 1, where the mapping $f : \mathbb{R}^{2p} \to \mathbb{R}^{2p}$ is defined as follows. We first convert the real vector $x_t$ into a complex vector $z_t \in \mathbb{C}^p$, where $z_t^{(j)} = x_t^{(2j-1)} + i\, x_t^{(2j)}$, for $j = 1, 2, \ldots, p$. We then apply a unitary matrix $U \in \mathbb{C}^{p\times p}$ to $z_t$, defining $z_t' = U z_t$. Next, we modulate the magnitude and phase of $z_t'$ with a scalar $q \in \mathbb{R}$ and a bias vector $\theta \in \mathbb{R}^p$ as follows:

$$z_t'' = \exp(i\theta) \odot (|z_t'| \odot \exp(iq \arg(z_t'))),$$

where $\odot$ denotes element-wise multiplication, $|z_t'|$ is the element-wise magnitude, and $\arg(z_t')$ is the element-wise phase of $z_t'$. We then compute $z_{t+1} = U^\star z_t''$, where $U^\star$ is the Hermitian transpose of $U$. Finally, we convert $z_{t+1}$ back to a real vector $x_{t+1} \in \mathbb{R}^{2p}$. The sequence starts from an initial vector $x_1 \in \mathbb{R}^{2p}$, normalized to unit length. The unitary matrix $U$ and bias vector $\theta$ are randomly initialized. The parameter $q$ controls the non-linearity applied to the phase of the transformed vector. In Figure 3 (right), we plot $\|u_t^\star - x_{t+1}\|^2$ against $t$ for $d = 4$, $q = 2$, and $k = k_{\exp}$. We observe convergence to zero, suggesting that our causal kernel descent method may generalize to more complex settings.

**Training $\mathcal{M}^n$.** In Theorem 1, the number of layers, $n$, is taken to infinity to obtain the estimate $u_t^\star$ of $x_{t+1}$. However, we experimentally demonstrate that fine-tuning the model $\mathcal{M}^n$ in equation 3 leads to a trained model that competes with the infinitely deep model $\mathcal{M}$. For this, we take $\mathcal{N} = \text{softmax}$ to build $\mathcal{M}^n$, and fine-tune the corresponding weights. Note that, for simplicity, we consider square matrices for the parameters, by completing the parameters with zeros. To emphasize the parameter dependency, we denote by $\mathcal{M}_{\theta_0}^n$ the corresponding initialization (which therefore satisfies Theorem 1). We take $d = 15$, $n = 6$, and consider instance (2) with randomly generated $\Omega$'s and $x_1$'s, for a dataset with $2^{12}$ elements, that we split into train, validation, and test sets with respective sizes of 60%, 20%, and 20% of the original dataset. We train the model using Adam (Kingma & Ba, 2014) on the Mean Squared Error (MSE) loss for next-token prediction on sequences of length $T = 100$, i.e., we minimize:

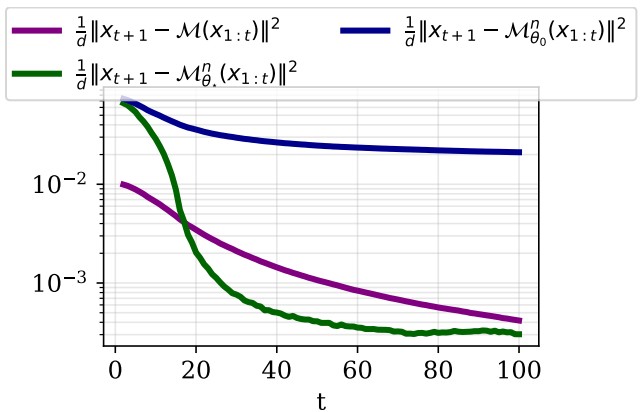

Figure 4: Errors $\frac{1}{d}\|\mathcal{G}(x_{1:t}) - x_{t+1}\|^2$ against $t$ for $\mathcal{G} \in \{\mathcal{M}, \mathcal{M}_{\theta_0}^n, \mathcal{M}_{\theta_\star}^n\}$. Results are averaged over the whole test set.

$$\ell(\theta) := \frac{1}{d} \sum_{t=1}^{T-1} \|\mathcal{M}_\theta^n(x_{1:t}) - x_{t+1}\|^2,$$

over the parameters $\theta$ of $\mathcal{M}^n$, starting from the initialization $\theta_0$. We train for 5000 epochs with early stopping. We denote by $\mathcal{M}_{\theta_\star}^n$ the corresponding model. We then examine how the error $\frac{1}{d}\|\mathcal{M}_{\theta_\star}^n(x_{1:t}) - x_{t+1}\|^2$ behaves with $t$. We find that not only does $\mathcal{M}_{\theta_\star}^n$ significantly outperform $\mathcal{M}_{\theta_0}^n$, but it also outperforms the infinite depth model $\mathcal{M}$ when $t \gtrsim 20$, as shown in Figure 5.

## CONCLUSION

In this paper, we took a step towards understanding the universality of Transformers for next-token prediction by considering sequences of the form $x_{t+1} = f(x_t)$ for some hidden variable $f$. We demonstrated in Theorem 1 that an explicitly constructed Transformer can accurately predict the next token $x_{t+1}$ as $t \to +\infty$, in specific cases where $f$ is linear or the sequence $(x_t)_{t\geq 1}$ is periodic. Our construction corresponds to the Transformer implementing causal kernel descent methods, which provably provide consistent estimates of the next token $x_{t+1}$ under the specific cases considered in this paper. Experimental results validated our theoretical findings and indicated that these methods can be extended to more general mappings $f$, paving the way for future investigations to generalize our theoretical results.

One current limitation of our approach is the deterministic aspect of the sequences considered in the paper. A possible extension of our work would be considering noisy dynamics of the form $x_{t+1} = f(x_t) + \varepsilon_t$ where the $\varepsilon_t$'s are i.i.d random variables. Another open problem concerns the ability of other sequential architectures such as RNNs or state-space models to learn in-context such autoregressive processes. These questions are left for future research.

## ACKNOWLEDGMENTS

The work of M. Sander and G. Peyré was supported by the French government under the management of Agence Nationale de la Recherche as part of the "Investissements d'avenir" program, reference ANR-19-P3IA-0001 (PRAIRIE 3IA Institute). The work of G. Peyré is supported by the European Research Council (ERC project WOLF). M. Sander thanks Francisco Andrade and Pierre Marion for fruitful discussions.

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

# A  PROOFS

## A.1  PROOF OF PROPOSITION 1

*Proof.* Let $n > 0$ be an integer.

We define positional encodings $p_t \in \mathbb{R}$ as

$$p_t = (-1)^t nt,$$

We concatenate the input embeddings and positional encodings and define

$$x_t^p = (x_t, p_t) \in \mathbb{R}^{d+1}.$$

We define the weight matrices $W_Q^{(h)}, W_K^{(h)}, W_V^{(h)}$ for each head $h = 1, 2$ as follows:

- $W_Q^{(1)} = [\mathbf{0}_{1 \times d}, \ 1], \quad W_K^{(1)} = [\mathbf{0}_{1 \times d}, \ -1], \quad W_V^{(1)} = \begin{pmatrix} I_d & \mathbf{0}_{d \times 1} \\ \mathbf{0}_{2d \times d} & \mathbf{0}_{2d \times 1} \end{pmatrix}.$

- $W_Q^{(2)} = [\mathbf{0}_{1 \times d}, \ 1], \quad W_K^{(2)} = [\mathbf{0}_{1 \times d}, \ 1], \quad W_V^{(2)} = \begin{pmatrix} \mathbf{0}_{d \times d} & \mathbf{0}_{d \times 1} \\ I_d & \mathbf{0}_{d \times 1} \\ \mathbf{0}_{d \times d} & \mathbf{0}_{d \times 1} \end{pmatrix}.$

With such constructions, one has

- $\langle W_Q^{(1)} x_t^p, W_K^{(1)} x_s^p \rangle = -p_s p_t$ and $W_V^{(1)} x_s^p = (x_s, 0, 0)$.

- $\langle W_Q^{(2)} x_t^p, W_K^{(2)} x_s^p \rangle = p_s p_t$ and $W_V^{(1)} x_s^p = (0, x_s, 0)$.

The corresponding attention scores are

$$\mathcal{A}_{t,s}^h = \frac{e^{(-1)^h p_s p_t}}{\sum_{\tau=1}^t e^{(-1)^h p_\tau p_t}} = \frac{1}{\sum_{\tau=1}^t e^{(-1)^h p_\tau p_t - (-1)^h p_s p_t}}.$$

When $h = 1$, one has that $(-1)^h p_s p_t = (-1)^{t-s+1} n^2 st$ is maximal and strictly positive when $s = t - 1$. Similarly, when $h = 2$, one has that $(-1)^h p_s p_t = (-1)^{t-s} n^2 st$ is maximal and strictly positive when $s = t$. Therefore, $\mathcal{A}_{t,s}^1 \to \delta_{s=t-1}$ and $\mathcal{A}_{t,s}^2 \to \delta_{s=t}$ as $n \to +\infty$.

We therefore consider the forward rule defined as, for $t \geq 1$:

$$\mathcal{F}^n(x_{0:T})_t = \sum_{s=1}^t \mathcal{A}_{t,s}^1 \cdot (x_s, 0, 0) + \sum_{s=1}^t \mathcal{A}_{t,s}^2 \cdot (0, x_s, 0).$$

As $n \to +\infty$, one has $\mathcal{F}^n(x_{0:T})_t \to (x_{t-1}, x_t, 0) \coloneqq \mathcal{F}(x_{0:T})_t$.

We then apply a feedforward map $\mathcal{G}$ on each token $(x_{t-1}, x_t, 0)$ defined as $g(a, b, c) = (\begin{pmatrix} a \\ 1_{a=0_d} \end{pmatrix}, \begin{pmatrix} b \\ 1 \end{pmatrix}, (1 - 1_{a=0_d})b, c)$ to obtain the desired augmented tokens $e_t$. The feedforward map $\mathcal{G}$ can be approximated with a sigmoid based perceptron $\mathcal{G}^n$ using the fact that $\frac{2}{1+e^{\|a\|n}} \to 1_{a=0_d}$ as $n \to +\infty$.

Denoting $\mathcal{T}_0$ the Transformer composed of the attention module $\mathcal{F}$ and the feedforward module $\mathcal{G}$ conludes the proof.

$\square$

## A.2 PROOF OF PROPOSITION 2

*Proof.* Recall that the matrix $A$ is defined as:

$$A_{t,s} = \begin{cases} k(x_t, x_s)1_{s \leq t}, & \text{if } k = k_{\text{id}} \\ k(x_t, x_s)1_{s \leq t} \text{ or } \frac{k(x_t, x_s)}{\sum_{\tau=1}^{t} k(x_t, x_\tau)}1_{s \leq t}, & \text{if } k = k_{\exp} \end{cases}.$$

We have in matrix notations:

$$u_{1:t}^n = (I_t - (I_t - \eta A)^n)A^{-1}(A - \text{diag}(A))x_{2:t+1}.$$

Importantly, when $A_{t,s} = k(x_t, x_s)$, for all $0 < \eta < \frac{2}{k(x_1, x_1)}$, one has $(I_t - \eta A)^n \to 0_{t \times t}$ exponentially fast. In this case, we therefore have $\eta^\star = \frac{2}{k(x_1, x_1)}$. Even more, when $\eta = \frac{1}{k(x_1, x_1)}$, the matrix $(I_t - \eta A)$ is nilpotent, and $(I_t - \eta A)^n = 0_{t \times t}$ for $n \geq t$.

When $A_{t,s} = \frac{k(x_t, x_s)}{\sum_{\tau=1}^{t} k(x_t, x_\tau)}$, for all $0 < \eta < 2$, one has $(I_t - \eta A)^n \to 0_{t \times t}$ exponentially fast. In this case, we therefore have $\eta^\star = 2$.

In both cases, when $0 < \eta < \eta^\star$, we have $\lim_{n \to +\infty} u_{1:t}^n = u_{1:t}^\star := A^{-1}(A - \text{diag}(A))x_{2:t+1}$. $\qquad\square$

## A.3 PROOF OF PROPOSITION 3

*Proof.* When $0 < \eta < \eta^\star$, at convergence, we have:

$$u_{1:t}^\star = x_{2:t+1} - A^{-1}\text{diag}(A)x_{2:t+1}.$$

Writing $z := A^{-1}\text{diag}(A)x_{2:t+1}$ and $\mu_s := \frac{z_s}{k(x_1, x_1)}$, we obtain the recursions on $\mu$:

$$\sum_{s=1}^{t} \mu_s k(x_s, x_t) = \frac{k(x_t, x_t)}{k(x_1, x_1)}x_{t+1} = W\varphi(x_t), \quad \forall t \geq 1, \tag{11}$$

where the second equality comes from the fact that $\|x_t\| = \|x_1\|$.

We have $u_t^\star = x_{t+1} - k(x_1, x_1)\mu_t$. $\qquad\square$

## A.4 PROOF OF PROPOSITION 4

*Proof.* We have

$$W_{t+1} = W_t + \mu_{t+1}\varphi(x_{t+1})^*.$$

Right multiplying by $\varphi(x_{t+1})$ gives

$$\mu_{t+1}k(x_{t+1}, x_{t+1}) = x_{t+2} - W_t\varphi(x_{t+1})$$

Therefore,

$$W_{t+1} = W_t(I - \frac{\varphi(x_{t+1})\varphi(x_{t+1})^*}{k(x_{t+1}, x_{t+1})}) + \frac{x_{t+2}}{k(x_{t+1}, x_{t+1})}\varphi(x_{t+1})^*.$$

Since

$$x_{t+2}\varphi(x_{t+1})^* = W\varphi(x_{t+1})\varphi(x_{t+1})^*,$$

we obtain

$$W_{t+1} = (W_t - W)(I - \frac{\varphi(x_{t+1})\varphi(x_{t+1})^*}{k(x_{t+1}, x_{t+1})}) + W$$

Since $W_1 = W\nu_1\nu_1^*$, it follows that $W_t - W = -WP_1P_2\cdots P_t$, where $P_s = (I - \nu_s\nu_s^*)$ is the orthogonal projection onto the subspace orthogonal to $\nu_s$. $\qquad\square$

## A.5 PROOF OF THEOREM 2

*Proof.* We first prove the second statement of the theorem, that is that $W_t - W \to 0$ exponentially fast for almost all $x_1$ and $W$.

Let's first fix $W$ and $x_1$. In what follows we denote $x := x_1$ to ease the notations.

Let $\Delta_t = W_t - W$. We have

$$\Delta_{t+1} = \Delta_t W^t (I - xx^\top) W^{-t}.$$

Unrolling, we obtain

$$\Delta_{t+1} = \Delta_1 (W(I - xx^\top))^t W^{-t} = -(W(I - xx^\top))^{t+1} W^{-t},$$

because $\Delta_1 = -W(I - xx^\top)$.

For fixed $W$ and $x$, let $y$ be an eigenvector of norm 1 of $W(I - xx^\top)$. There exists $\lambda$ such that

$$W(I - xx^\top)y = \lambda y.$$

If $|\lambda| = 1$, then because $W$ preserves the norm, it follows that

$$\|(I - xx^\top)y\|^2 = \|y\|^2 = 1.$$

Developing, we get

$$\|x\langle x, y\rangle\|^2 = 0.$$

But since

$$\|x\langle x, y\rangle\|^2 = \langle x, y\rangle^2,$$

we must have $\langle x, y\rangle = 0$, and therefore $Wy = \lambda y$, so that $\lambda$ is also an eigenvalue of $W$.

Therefore, to show that for almost all $x$ and $W$, $\rho(W(I - xx^\top)) < 1$, it suffices to show that for almost all $y \in V(W)$:

$$\langle x, y\rangle \neq 0,$$

where $V(W)$ denotes the eigen vectors of unit norms of $W$.

Let $DO(d)$ be the subset of $O(d)$ of orthogonal matrices with distinct eigenvalues and let $W \in DO(d)$.

Let $\lambda_1, \cdots, \lambda_d$ be the eigenvalues of $W$ and $b_1, \cdots, b_d$ be $d$ corresponding eigenvectors. Let $y \in V(W)$, and $\lambda$ be the corresponding eigenvalue.

Writing $y = \sum_{i=1}^d y_i b_i$, one has on the one hand that

$$Wy = \lambda y$$

and on the other hand

$$Wy = \sum_{i=1}^d y_i \lambda_i b_i.$$

Identifying the coefficients gives $\lambda = \lambda_i$ whenever $y_i \neq 0$. Since the $\lambda_i$'s are all distinct, necessarily exactly one $y_i$ is non-zero, and $y \in \mathbb{C}b_i$.

Therefore, if $\langle x, y\rangle = 0$, then $x \in \cup_{i=1}^d (\mathbb{C}b_i)^\perp$. However, for almost all $x \in S^{d-1}$, $x \notin \cup_{i=1}^d (\mathbb{C}b_i)^\perp$. Therefore, for almost all $W \in DO(d)$ and $x \in S^{d-1}$,

$$\langle x, y\rangle \neq 0.$$

Since for almost all $W \in O(d)$, $W \in DO(d)$, we conclude that $\rho(W(I - xx^\top)) < 1$ for almost all $x \in S^{d-1}$ and $W \in O(d)$. As a consequence, for almost all $x \in S^{d-1}$ and $W \in O(d)$, $W_t - W \to 0$ as $t \to +\infty$. It already implies that, for almost all $x$ and $W$, $\mu_t x_t^\top \to 0$. Therefore, $\text{Tr}(\mu_t x_t^\top x_t \mu_t^\top) \to 0$. But since $x_t^\top x_t = 1$ and $\text{Tr}(\mu_t \mu_t^\top) = \|\mu_t\|$, it gives $\mu_t \to 0$, and therefore $u_t^\star - x_{t+1} \to 0$ for almost all $x_1$ and $W$.

However, to fully prove the first statement of the theorem, that is that, **for all** $x_1$ and $W$, $u_t^\star - x_{t+1} \to 0$, we follow the exact same demonstration as for Theorem 3 (corresponding to the next proof in Appendix A.6), simply by replacing exp by id in equation 12. $\square$

## A.6 PROOF OF THEOREM 3

*Proof.* As mentioned in the proof sketch, we consider the subset of $\mathcal{H}$ of linear combinations of the $\nu_s$:

$$x \mapsto \sum_{s=1}^{\tau} a_s \nu_s, \text{ for } a_s \in \mathbb{R}^d, \tau \geq 1.$$

This subset is a pre-Hilbert space under the inner product inherited from $\mathcal{H}$. By completing this space with respect to the induced norm from $\mathcal{H}$, we obtain a new Hilbert space $\mathcal{H}'$

We first show that $P_t P_{t-1} \cdots P_1 \to 0$ strongly in $\mathcal{H}'$, using a characterization from Kwapień & Mycielski (2001).

A sequence $(\nu_s)_{s \geq 1}$ of unit vector for which $P_t P_{t-1} \cdots P_1 \to 0$ strongly is referred to as *effective*.

The specificities of our case are twofold: first, $\nu_s \in \mathcal{H}'$, which is potentially of infinite dimension, and second, the vectors $\nu_s$ follow an autoregressive relation. In our case, we can still show that the sequence $(\nu_s)_{s \geq 1}$ is effective in $\mathcal{H}'$.

Note that because $\Omega \in O(d)$, one has for any positive integers $t, s, r$ that $\langle \nu_{s+r}, \nu_{t+r} \rangle = \langle \nu_s, \nu_t \rangle$. Such sequence is called *stationary*.

Bochner's theorem states that there is a measure $\sigma$ on the unit circle $S^1$–called *spectral measure*–such that, for all $t \geq 1$,

$$a_t := \langle \nu_{t+1}, \nu_1 \rangle = \int_{S^1} z^t \, d\sigma(z).$$

We are going to use the following characterization from Kwapień & Mycielski (2001); Rainis Haller (2005).

**Theorem** (Effectiveness of stationary sequences (Kwapień & Mycielski, 2001)). *A stationary sequence of unit vectors which is linearly dense in a Hilbert space is effective if and only if its spectral measure either coincides with the normalized Lebesgue measure or is singular with respect to the Lebesgue measure.*

Now, let $\Omega$ be an orthogonal matrix. One has

$$a_t = \langle \nu_{t+1}, \nu_1 \rangle = \frac{1}{k(x_1, x_1)} k(x_{t+1}, x_1) = \frac{1}{k(x_1, x_1)} \exp\left(\langle x_1, \Omega^t x_1 \rangle\right).$$

Since $\Omega$ is an orthogonal matrix, it can be diagonalized via rotations with angles $\theta_1, \theta_2, \ldots, \theta_p$.

We therefore can write $\langle x_1, \Omega^t x_1 \rangle$ as

$$\langle x_1, \Omega^t x_1 \rangle = \sum_{i=1}^{p} (x_{1(2i-1)}^2 + x_{1(2i)}^2) \cos(\theta_i t) + \sum_{i=2p+1}^{m} x_{1i}^2 - \sum_{i=m+1}^{d} x_{1i}^2,$$

for some $p$ and $m$. Therefore, one can write

$$a_t = g(t(\theta_1, \cdots, \theta_p)),$$

where $g$ is defined as

$$g(y_1, \cdots, y_p) := \exp\left(-1 + \sum_{i=1}^{p} (x_{1(2i-1)}^2 + x_{1(2i)}^2) \cos(y_i) + \sum_{i=p+1}^{m} x_{1i}^2 - \sum_{i=m+1}^{d} x_{1i}^2\right). \quad (12)$$

The function $g$ is periodic in each variable $y_i$ with period $2\pi$. Since $g$ is also $C^1$, this allows us to represent $g$ as a Fourier series:

$$g(y) = \sum_{k \in \mathbb{Z}^p} c_k e^{i\langle y, k \rangle}.$$

Therefore,

$$a_t = \sum_{k \in \mathbb{Z}^p} c_k e^{it\langle \theta, k \rangle}.$$

Therefore, the spectral measure satisfies

$$\sigma = \sum_{k \in \mathbb{Z}^p} c_k \delta_{\langle \theta, k \rangle}.$$

Note that this necessarily implies that $c_k \geq 0$ by Bochner's theorem. We are now going to prove that $\sigma$ is singular with respect to the Lebesgue measure.

We recall that two positive measures $\alpha$ and $\beta$ defined on a measurable space $C$ are singular if there exist two disjoint measurable sets $A, B$ such that $A \cup B = C$, such that $\alpha$ is zero on all measurable subsets of $B$, while $\nu$ is zero on all measurable subsets of $A$. Here, $\alpha = \sigma$, and $\beta$ is the Lebesgue measure on $C = S^1$.

We have

$$A = \cup_{k \in \mathbb{Z}^p} \{\langle \theta, k \rangle\}.$$

$A$ is measurable and the Lebesgue measure $\beta$ is zero on $A$, while by definition, $\sigma$ is zero on all measurable subsets of $B := S - A$.

Therefore, $\sigma$ is singular with respect to the Lebesgue measure, which, using Theorem 2 from Kwapień & Mycielski (2001) in the Hilbert space $\mathcal{H}'$ shows that the sequence $(\nu_s)_{s \geq 1}$ is effective in $\mathcal{H}'$.

Now, we have, by taking the adjoint $^*$ of $W_t - W$:

$$W_t^* - W^* = -P_t \cdots P_1 W^*.$$

We recall that we identify $W$ with $W_{\mathcal{H}'}$ so that $W : \mathcal{H}' \to \mathbb{R}^d$ and $W^* : \mathbb{R}^d \to \mathcal{H}'$. One has for all $x \in \mathbb{R}^d$

$$(W_t^* - W^*)x = \sum_{s=1}^{t} \varphi(x_s)\langle z_s, x \rangle - W^* x = -P_t \cdots P_1 W^* x \to 0$$

because $W^* x \in \mathcal{H}'$.

Therefore, for all $x \in \mathbb{R}^d$, $\varphi(x_t)\langle z_t, x \rangle \to 0$ in $\mathcal{H}'$. Because $\|\varphi(x_t)\|_{\mathcal{H}'} = k(x_1, x_1) > 0$, we necessarily have $\langle z_t, x \rangle \to 0$ for all $x \in \mathbb{R}^d$. This is equivalent to $z_t \to 0$, and therefore $u_t^\star - x_{t+1} \to 0$.

$\square$

### A.7 PROOF OF THEOREM 4

*Proof.* Let $t_p$ denotes the period of $(x_t)_{t \geq 1}$. We define

$$\Pi := P_1 \cdots P_{t_p}.$$

One has $\sup_{\nu \in \mathcal{H}', \|\nu\|=1} \|\Pi \nu\| < 1$ (see Rainis Haller (2005), page 2). Therefore, $\Pi^m \to 0$ as $m \to +\infty$.

As such, for any $x$ such that $\varphi(x) \in \mathcal{H}'$, $W_t \varphi(x) \to W\varphi(x)$ exponentially fast.

In particular, for $x = x_1$, we get that

$$\sum_s \mu_s k(x_s, x_1)$$

converges exponentially fast. Since $k(x_s, x_1) > \exp(-1)$, it implies that $\mu_s \to 0$ exponentially fast. $\square$

### A.8 PROOF OF PROPOSITION 5

*Proof.* **Building positional encodings and parameters.**

Similarly to the previous proof, we consider positional encodings $p_t^1$ and $p_t^2 \in \mathbb{R}$ as

$$p_t^1 = \delta_{s=1}, \quad p_t^2 = 1.$$

We concatenate the input embeddings and positional encodings and define

$$e_{t,p} = (x_{t-1}, p_t^1, x_t, p_t^2, x_t, u_t) = (x_{t-1}, 0, x_t, 1, x_t, u_t) \in \mathbb{R}^{4d+2}$$

if $t > 1$ and

$$e_{1,p} = (0_d, p_1^1, x_1, p_1^2, 0_d, u_1) = (0_d, 1, x_1, 1, 0_d, u_1).$$

We define the weight matrices $W_Q^{(h)}, W_K^{(h)}, W_V^{(h)}$ for each head $h = 1, 2$ as follows:

- $W_Q^{(1)} = W_K^{(1)} = [\mathbf{0}_{d\times d+1}, I_d, \mathbf{0}_{d\times 2d+1}]$, and

$$W_V^{(1)} = -\eta \begin{pmatrix} \mathbf{0}_{d+1\times d+1} & \mathbf{0}_{d+1\times d+1} & \mathbf{0}_{d+1\times d} & \mathbf{0}_{d+1\times d} \\ \mathbf{0}_{d+1\times d+1} & \mathbf{0}_{d+1\times d+1} & \mathbf{0}_{d+1\times d} & \mathbf{0}_{d+1\times d} \\ \mathbf{0}_{d\times d+1} & \mathbf{0}_{d\times d+1} & \mathbf{0}_{d\times d} & \mathbf{0}_{d\times d} \\ \mathbf{0}_{d\times d+1} & \mathbf{0}_{d\times d+1} & \mathbf{0}_{d\times d} & I_d \end{pmatrix}.$$

- $W_Q^{(2)} = [\mathbf{0}_{d+1\times d+1}, I_{d+1}, \mathbf{0}_{d+1\times 2d}]$, $W_K^{(2)} = [I_{d+1}, \mathbf{0}_{d+1\times d+1}, \mathbf{0}_{d+1\times 2d}]$, $W_V^{(2)} =$

$$\eta \begin{pmatrix} \mathbf{0}_{d+1\times d+1} & \mathbf{0}_{d+1\times d+1} & \mathbf{0}_{d+1\times d} & \mathbf{0}_{d+1\times d} \\ \mathbf{0}_{d+1\times d+1} & \mathbf{0}_{d+1\times d+1} & \mathbf{0}_{d+1\times d} & \mathbf{0}_{d+1\times d} \\ \mathbf{0}_{d\times d+1} & \mathbf{0}_{d\times d+1} & \mathbf{0}_{d\times d} & \mathbf{0}_{d\times d} \\ \mathbf{0}_{d\times d+1} & \mathbf{0}_{d\times d+1} & I_d & \mathbf{0}_{d\times d} \end{pmatrix}.$$

With such constructions, one has

- $\langle W_Q^{(1)} e_{t,p}, W_K^{(1)} e_{s,p} \rangle = \langle x_t, x_s \rangle$ and $W_V^{(1)} e_{s,p} = -\eta(0_{d+1}, 0_{d+1}, 0_d, u_s)$.

- $\langle W_Q^{(2)} e_{t,p}, W_K^{(2)} e_{s,p} \rangle = \langle x_t, x_{s-1} \rangle + p_s^1 p_t^2 = \langle x_t, x_{s-1} \rangle + \delta_{s=1}$ and $W_V^{(2)} e_{s,p} = \eta(0_{d+1}, 0_{d+1}, 0_d, x_s)$ if $s > 1$ and $W_V^{(2)} e_{1,p} = (0_{d+1}, 0_{d+1}, 0_d, 0_d)$.

**Implementing equation 7 when $A_{t,s} = k(x_t, x_s)$.**

Let $\mathcal{T}$ be a one-layer, two-head Transformer with the parameters above (with $\mathcal{N} = $ id of $\mathcal{N} = \exp$). Its forward rule is defined as, starting from $e_{t,p}^0 = e_{t,p}$:

$$e_{t,p}^{k+1} = e_{t,p}^k - \eta \sum_{s=1}^t \mathcal{A}_{s,t}^1 (0_{d+1}, 0_{d+1}, 0_d, u_s^k) + \eta \sum_{s=1}^t \mathcal{A}_{s,t}^2 (0_{d+1}, 0_{d+1}, 0_d, x_s).$$

Therefore, the $(3d + 2)$ first coordinates of $e_{t,p}^k$ are not modified and the $d$ last coordinates $u_t^k$ are updated as

$$u_t^{k+1} = u_t^k - \eta \sum_{s=1}^t k(x_t, x_s) u_s^k + \eta \sum_{s=2}^t k(x_t, x_{s-1}) x_s \tag{13}$$

The second term in equation 13 reads

$$\sum_{s=2}^t k(x_t, x_{s-1}) x_s = \sum_{s=1}^{t-1} k(x_t, x_s) x_{s+1}.$$

Therefore, equation 13 reads

$$u_t^{k+1} = u_t^k - \eta \sum_{s=1}^t k(x_t, x_s)(u_s^k - 1_{s<t} x_{s+1}),$$

which corresponds exactly to equation 7.

**Implementing equation 7 when $A_{t,s} = \frac{k(x_t, x_s)}{\sum_{\tau=1}^t k(x_t, x_\tau)}$.**

Let $\mathcal{T}$ be a $\mathcal{N} = $ softmax-based, one-layer, two-head Transformer with the parameters above. Its forward rule is defined as

$$e_{t,p}^{k+1} = e_{t,p}^k - \eta \sum_{s=1}^t \mathcal{A}_{s,t}^1 (0_{d+1}, 0_{d+1}, 0_d, u_s^k) + \eta \sum_{s=1}^t \mathcal{A}_{s,t}^2 (0_{d+1}, 0_{d+1}, 0_d, x_s).$$

Therefore, the $(3d + 2)$ first coordinates of $e_{t,p}$ are not modified and the $d$ last coordinates $u_t$ are updated as

$$u_t^{k+1} = u_t^k - \eta \sum_{s=1}^{t} \frac{e^{\langle x_t, x_s \rangle}}{\sum_{\tau=1}^{t} e^{\langle x_t, x_\tau \rangle}} u_s^k + \eta \sum_{s=2}^{t} \frac{e^{\langle x_t, x_{s-1} \rangle + \delta_{s=1}}}{\sum_{\tau=1}^{t} e^{\langle x_t, x_{\tau-1} \rangle + \delta_{\tau=1}}} x_s. \tag{14}$$

Because $\|x_t\| = 1$ for all $t \geq 1$ and $x_0 = 0$, one has

$$\sum_{\tau=1}^{t} e^{\langle x_t, x_{\tau-1} \rangle + \delta_{\tau=1}} = \sum_{\tau=1}^{t} e^{\langle x_t, x_\tau \rangle}$$

so that the second term in equation 14 reads

$$\sum_{s=2}^{t} \frac{e^{\langle x_t, x_{s-1} \rangle + \delta_{s=1}}}{\sum_{\tau=1}^{t} e^{\langle x_t, x_{\tau-1} \rangle + \delta_{\tau=1}}} x_s = \sum_{s=2}^{t} \frac{e^{\langle x_t, x_{s-1} \rangle}}{\sum_{\tau=1}^{t} e^{\langle x_t, x_\tau \rangle}} x_s = \sum_{s=1}^{t-1} \frac{e^{\langle x_t, x_s \rangle}}{\sum_{\tau=1}^{t} e^{\langle x_t, x_\tau \rangle}} x_{s+1}.$$

Therefore, equation 14 reads

$$u_t^{k+1} = u_t^k - \eta \sum_{s=1}^{t} \frac{e^{\langle x_t, x_s \rangle}}{\sum_{\tau=1}^{t} e^{\langle x_t, x_\tau \rangle}} (u_s^k - 1_{s<t} x_{s+1}),$$

which corresponds exactly to equation 7.

$\square$

## A.9 PROOF OF THEOREM 1

*Proof.* We first use Proposition 5. For any $0 < \eta < \eta^\star$, where $\eta^\star$ of defined in Proposition 2 and for each configuration of equation 7, there exists an attention-only, one-layer, two-head causal Transformer $\mathcal{T}$ such that, for any autoregressive sequence $(x_t)_{t \geq 1}$ generated according to Assumption 1, defining

$$e_t^k := (x_{t-1}, 0, x_t, 1, x_t, u_t^k)$$

for $t > 1$ and

$$e_1^k := (0_d, 1, x_t, 1, 0_d, u_t^k),$$

we have that, for any $n$, $e_{1:t}^{1:n}$ solves equation 3 if and only if $u_{1:t}^{1:n}$ solves equation 7.

Thanks to Proposition 2, we know that $u_t^n \to_{n \to +\infty} u_t^\star$, and define $\mathcal{M}(x_{1:t}) := u_t^\star$. Then:

- When $A_{t,s} = \langle x_t, x_s \rangle$, taking $\mathcal{N} = \mathrm{id}$ in the construction of $\mathcal{T}$ in the proof of Proposition 5, and under instance (1), we have $\mathcal{M}(x_{1:t}) - x_{t+1} \to 0$, with exponential speed for almost all $x_1$ and $W$, thanks to Theorem 2.

- When $A_{t,s} = e^{\langle x_t, x_s \rangle}$ (resp. $A_{t,s} = \frac{e^{\langle x_t, x_s \rangle}}{\sum_{\tau=1}^{t} e^{\langle x_t, x_\tau \rangle}}$), taking $\mathcal{N} = \exp$ (resp. $\mathcal{N} = \mathrm{softmax}$) in the construction of $\mathcal{T}$ in the proof of Proposition 5, and under instance (2), we have $\mathcal{M}(x_{1:t}) - x_{t+1} \to 0$ thanks to Theorem 3.

- When $A_{t,s} = e^{\langle x_t, x_s \rangle}$ (resp. $A_{t,s} = \frac{e^{\langle x_t, x_s \rangle}}{\sum_{\tau=1}^{t} e^{\langle x_t, x_\tau \rangle}}$), taking $\mathcal{N} = \exp$ (resp. $\mathcal{N} = \mathrm{softmax}$) in the construction of $\mathcal{T}$ in the proof of Proposition 5, and under instance (3), we have $\mathcal{M}(x_{1:t}) - x_{t+1} \to 0$, with exponential speed, thanks to Theorem 4.

The last statement of Theorem 1 follows from Proposition 2. When $\mathcal{N} = \mathrm{id}$ or $\mathcal{N} = \exp$, by choosing $\eta = \frac{1}{k(x_1, x_1)}$, we have that $\mathcal{M}^n(x_{1:t}) = u_t^n = u_t^\star = \mathcal{M}(x_{1:t})$ whenever $n \geq t$.

When $\mathcal{N} = \mathrm{softmax}$ however, choosing for instance $\eta = 1$ in the proof of Proposition A.2, we get $u_{1:t}^n - u_{1:t}^\star = (I_t - A)^n u_{1:t}^\star$. Since $\rho(I_t - A) \leq 1 - \frac{1}{t}$, it is reasonable to conjecture that $\|u_t^n - u_t^\star\|$ is of order $(1 - \frac{1}{t})^n$. In this case, it is sufficient to impose $\frac{t}{n} \to 0$ as $t, n \to +\infty$ to have $\varepsilon_1(n, t) := \|u_t^n - u_t^\star\| \to 0$ as $n, t \to +\infty$. $\square$

## B  ADDITIONAL RESULTS

**Dual interpretation.**   We provide a dual interpretation for $\mu$ defined in Proposition 3. We conveniently write equation 5 as $\min_W \|\Phi(W) - x_{2:T+1}\|^2$ where for $\mu$ in $\mathbb{R}^{d \times T-1}$, $\Phi(W) := (W\varphi(x_t))_t$ and $\Phi^\top(\mu) = \sum_t \mu_t \varphi(\mu_t)^\top$. Gradient flow on $E$ reads:

$$\dot{W} = -\nabla E(W) = -[\Phi^\top(\Phi(W) - x_{2:T+1})].$$

At optimality, one has $W = \Phi^T(\mu)$ for some $\mu \in \mathbb{R}^{d \times T-1}$ ("kernel trick"), so that we consider the energy $F(\mu) = E(\Phi^T(\mu))$. Gradient flow on $F$ reads: $\dot{\mu} = -K[K\mu - x_{2:T+1}]$ where $K := \Phi\Phi^\top :$ $\mu \mapsto \left(\sum_s k(x_s, x_t)\mu_s\right)_t$. Since $K$ is positive, we instead consider the equivalent flow

$$\dot{\mu} = -[K\mu - x_{2:T+1}].$$

Making this flow causal by replacing $K$ with its masked counterpart $A$ recovers to the results of the previous paragraph, since, at convergence, $\mu$ satisfies equation 8.

