# OpenReview forum: "Towards Understanding the Universality of Transformers for Next-Token Prediction"
_ICLR.cc/2025/Conference — ICLR 2025 Poster_

### Official Review · Reviewer_mAs7 · 2024-10-29

**Soundness:** 3
**Presentation:** 2
**Contribution:** 3
**Rating:** 5
**Confidence:** 3

**Summary:**

The paper aims to advance our understanding of causal Transformers for next-token prediction in autoregressive settings by showing how to construct networks that can solve the next-token prediction in-context learning task.

**Strengths:**

**Despite not being very familiar with kernel methods and learning theory (but somewhat familiar with Transformers and in-context learning), I believe that the authors' contributions are relevant and interesting, though I cannot assess their (learning theoretical) details.**

Particular, I find the paper is strong in the following aspects:
- crisp and clear setup in the abstract
- intuitive Fig 1 (for the setting, not for the results)
- very well summarized contributions list
- the results seem to be strong (I am uncertain about how realistic the setting is)

Nonetheless, as I will detail below, **I think the presentation needs to be significantly improved**.

**Weaknesses:**

### Major points
- the paper is very dense, even though the authors could have used an additional 10th page. I'd suggest more explanations for the result, and less formulas in the main text
- the intuition seems to be missing (for me) as to why you need the augmented tokens. What do they mean?
- Figure 2 needs to be explained; the one-sentence reference in L342 does not explain (for me, at least) what all the quantities in the figure are and how I should think about them).
- **It is unclear to me how a Transformer can implement Eq (7) if one problem is with the non-causal descent.** You still need to modify the training method, right? Please elaborate how this works/correct me if I am wrong (I see how you can construct the Transformer, but I do not see how the Transformer itself can account for the change in the descent method). If this point is about the "meta-optimization" during solving the in-context learning task, then please say so explicitly
- What is the rationale behind the construction of $f$ in the _"More Complex Iterations"_ paragraph?

### Minor points
- please use equation numbers for the main equations
- please define all quantities before you use them
- Definition 1: what is $S^{d-1}$?
- Definition 2: what is $O(\cdot)$?
- Definitions 1 and 2 would be better suited as assumptions
- L206: please explain what $e$ is, what the indices stand for, and why you add multiple beginning-of-sequence tokens and tokens of "1".
- Proposition 1: what is $\mathcal{N}?$
- L246: what makes these constructions explicit? As far as I understand, you specify the dimensions, but that leaves many degrees of freedom (ie, all the elements can be of any value)?
	- From Prop. 5 and checking A.8, I get a sense what these matrices would be. Thus, as you have a contruction, please refer to this fact to the reader, otherwise, calling a matrix explicit without saying that you can calculate all the values can be misleading.
- L286: do you mean the update of $u_{t}$ depends on $x_{1:t}?$

**Questions:**

- L178: why is your setup better reflective of how LLMs are trained? Please elaborate.
- L194: Why does the unit-norm of $x$ imply that the kernel values for $k(x_{i},x_{i})$ are same for all $i$?
- Eq (3): why do you need to model the projection (in this exact way of picking the last $d$ coordinates)?

---

> ### Author Response · Authors · 2024-11-23
>
> Thank you for reviewing our paper and for the insightful remarks. Below, we address each of your points in detail, hoping to convince the reviewer to increase their score.
>
> > The paper is very dense, even though the authors could have used an additional 10th page. I'd suggest more explanations for the results and fewer formulas in the main text.
>
> Thank you for pointing this out. While we adhered to the recommended 9-page limit, we will make use of the 10th page in the final version to improve the readability of the paper. Specifically, we will add more explanations to clarify key results and streamline the presentation of the formulas.
>
> > The intuition seems to be missing (for me) as to why you need the augmented tokens. What do they mean?
>
> The intuition behind augmented tokens is that they enable the model to directly access two successive elements in the sequence. Importantly, while prior works, such as von Oswald et al. (2023a, 2023b), directly worked on augmented tokens, our contribution lies in showing how they can be computed using positional encoding-based attention.
>
> Additionally, von Oswald et al. (2023b) provide experimental evidence that such tokens are formed numerically in real-world models, validating their practical relevance.
>
> > Figure 2 needs to be explained; the one-sentence reference in L342 does not explain (for me, at least) what all the quantities in the figure are and how I should think about them.
>
> We will provide a more detailed explanation for Figure 2 in the revised version. Briefly, each point in the figure corresponds to an iterate $\nu_t$, defined as $\nu_t = P_1 \cdots P_t \nu$. The figure illustrates the effect of iterating projections on a vector $\nu$ in finite dimensions, demonstrating how the vector evolves under these transformations.
>
> > It is unclear to me how a Transformer can implement Eq. (7) if one problem is with the non-causal descent. You still need to modify the training method, right? Please elaborate on how this works/correct me if I am wrong (I see how you can construct the Transformer, but I do not see how the Transformer itself can account for the change in the descent method). If this point is about the "meta-optimization" during solving the in-context learning task, then please say so explicitly.
>
> There is no training method in our paper. Instead, we construct a model that directly implements the causal kernel descent method in its forward pass. As the reviewer notes, this is related to meta-optimization: our key contribution is showing how the Transformer architecture can implement an optimization algorithm.
>
> To clarify, we explicitly construct a Transformer that performs causal kernel descent and prove its convergence for specific choices of sequences. We will add a remark in the final version to explicitly relate this to meta-optimization, even though the term may not fully apply in our causal setting.
>
> > What is the rationale behind the construction in the "More Complex Iterations" paragraph?
>
> We will expand on this explanation in the revised version. The key idea is to apply "non-linear rotations" to 2-dimensional chunks of the input. Each of these rotations is parameterized by an angle $\theta$ and a scalar $q$, and defined as $F_{\theta, q}(z) := R_{\theta} z^q$, where $R_{\theta}$ is a 2D rotation of angle $\theta$. Our construction generalizes this by randomly rotating the input before applying these transformations.

---

> > ### Author Response · Authors · 2024-11-23
> >
> > > Please use equation numbers for the main equations.
> >
> > We will add equation numbers in the final version. Thank you for the suggestion.
> >
> > > Please define all quantities before you use them.
> >
> > There is a notation paragraph (l.157) where all symbols are defined. We will make this reference more explicit.
> >
> > > Definition 1: What is $S^{d-1}$?
> >
> > $S^{d-1}$ refers to the $d-1$-dimensional unit sphere. This is defined in the notation paragraph.
> >
> > > Definition 2: What is $O(d)$?
> >
> > $O(d)$ is the orthogonal group in $d$ dimensions, defined in the notation paragraph.
> >
> > > Definitions 1 and 2 would be better suited as assumptions.
> >
> > We agree and will label them as assumptions in the revised version.
> >
> > > L206: Please explain what is being defined, what the indices stand for, and why you add multiple beginning-of-sequence tokens and tokens of "1".
> >
> > The notation $:=$ indicates that we are defining augmented tokens at this point. The augmented tokens incorporate standard beginning-of-sequence tokens, while the "1" corresponds to concatenated positional encodings, which are technically required in our proof. We formally prove that these tokens can be computed within a Transformer.
> >
> > > Proposition 1: What is $\mathcal{N}$?
> >
> > $\mathcal{N}$ is the attention normalization operator, defined earlier in l.101.
> >
> > > L246: What makes these constructions explicit? As far as I understand, you specify the dimensions, but that leaves many degrees of freedom (i.e., all the elements can take arbitrary values).
> >
> > The constructions are explicit because we provide their exact expressions in Appendix A.8. To avoid confusion, we will move these details to the main text, leveraging the additional page.
> >
> > > L286: Do you mean the update of $u_t$ depends on $x_{1:t}$?
> >
> > No, this is not a typo. The update of $u_t$ depends on $x_{1:T}$, which is problematic for causal Transformers. We explicitly show how this dependency is addressed in our construction.
> >
> > ---
> >
> > **Questions**
> >
> > > L178: Why is your setup better reflective of how LLMs are trained? Please elaborate.
> >
> > This paragraph states that our setup better reflects the nature of sentences used to train large language models (LLMs). In prior literature on in-context learning, tokens are typically modeled as independent and identically distributed (i.i.d.) pairs $(x_i, y_i)$. However, this does not capture the autoregressive dependencies present in natural language.
> >
> > Our proposed model incorporates autoregressive relationships, aligning more closely with the data LLMs encounter during training.
> >
> > > L194: Why does the unit-norm of $x$ imply that the kernel values for $k(x_i, x_i)$ are the same for all $i$?
> >
> > This follows because $k(x_i, x_i)$ is a function of $\|x_i\|$ alone. When $\|x_i\| = 1$, we have $k(x_i, x_i) = k(x_1, x_1) = 1$.
> >
> > > Eq. (3): Why do you need to model the projection in this exact way, by picking the last \(d\) coordinates?
> >
> > The augmented tokens are in $\mathbb{R}^{4d+2}$, but the output must be in $\mathbb{R}^d$. A projection onto $d$ coordinates is the simplest way to achieve this. We specifically track $u_t$ in the last $d$ coordinates for this purpose.
> >
> > ---
> >
> > We strongly hope the reviewer will consider increasing their score in light of our rebuttal.

---

> ### Comment · Reviewer_mAs7 · 2024-11-23
>
> Thank you for your explanations! Now I better understand the contributions of the paper, and will consider raising my score, but only if there will be a revision of the manuscript during the rebuttal period. You have agreed to carry out many changes, and I would only be confident if I could read the new version, and ask questions in case further clarifications are required.

---

> > ### Author Response · Authors · 2024-11-24
> >
> > Thank you very much for considering increasing your score. We have provided a revision of the manuscript, where all the changes we committed to make are incorporated and appear in purple. Please let us know if this satisfies you or if further clarifications are required.
> >
> > Thank you again

---

> > > ### Author Response · Authors · 2024-11-26
> > >
> > > Dear reviewer,
> > >
> > > We have until tomorrow to update the pdf. Please let us know if the modifications we made satisfy you and if further clarifications are needed.
> > >
> > > Thanks a lot

---

> > > > ### Comment · Reviewer_mAs7 · 2024-11-28
> > > >
> > > > Dear Authors,
> > > >
> > > > Sorry, I missed your message on the 24th. I appreciate the changes, which have made the paper easier to follow. I'd still add more explanation to the caption of Fig. 2. But as this is a small remaining fix you already promised to do, I have increased my score.

---

> > > > > ### Author Response · Authors · 2024-11-28
> > > > >
> > > > > Dear reviewer,
> > > > >
> > > > > Thank you for your response and for increasing your score.
> > > > >
> > > > > Best,
> > > > >
> > > > > The authors

---

### Official Review · Reviewer_6GYT · 2024-11-04

**Soundness:** 3
**Presentation:** 3
**Contribution:** 3
**Rating:** 6
**Confidence:** 2

**Summary:**

This paper theoretically studies the universality of Transformers for next-token prediction. The authors specifically consider the sequences of the form $x_{t+1} = f(x_t)$. In particular, they theoretically analyze two types of instances:  for linear $f$ and periodic sequences (exponential kernel). They construct augmented tokens and show that an explicitly constructed Transformer can learn in context to accurately predict the next token asymptotically, through a causal kernel descent method.

**Strengths:**

1. Very mathematically sound and interesting. The universality of this next-token predictability is a strong result.

2. Causal kernel descent is a very intriguing framework to analyze sequence models in context prediction, taking the important aspect of masking out the future into account.

**Weaknesses:**

1. The theoretical setting is limited to when $f$ is linear and when the sequence is periodic, with each point in the context being on the unit sphere. Neither cases seem too complicated in the first place, and so the "universality" does not seem well justified in the context of Theorem 1. Perhaps the authors can justify the reduction to these sub-classes of functions more explicitly.

2. Both the process of generating the augmented token $e_t^0$ from $x_{0:t}$, as well as the causal kernel descent iteration theoretically involves an infinitely deep Transformer as $n\to\infty$. It might be more informative to discuss the theoretical implications of finite depth.

3. The universality of next token predictability result is asymptotic in the context length $t$. Although the convergence is fast enough (exponential), most empirical observations are in the non-asymptotic regime in terms of context length. It might be good to explicitly provide analysis on this.

**Questions:**

1. 250 typo $\epsilon_1(n,t) = \mathcal{M}^n(x_{1:t}) - \mathcal{M}(x:1,t)$?

2. In the experiment section, how did you compare fine-tuning $\mathcal{M}^n$ with infinitely deep model? Did you just take $\mathcal{M}$ to be the ground truth? (Figure 5 is missing)

---

> ### Author Response · Authors · 2024-11-23
>
> Thank you very much for your review and for the nice comments. Please see our responses to your concerns below, hoping they can convince the reviewer to increase their score.
>
> > The theoretical setting is limited to when $f$ is linear and when the sequence is periodic, with each point in the context being on the unit sphere. Neither case seems too complicated in the first place, and so the "universality" does not seem well justified in the context of Theorem 1. Perhaps the authors can justify the reduction to these sub-classes of functions more explicitly.
>
> We would like to emphasize that supposing linear dependency in the data is common in the in-context learning literature; see the related background section, l. 119. The difficulty in our work lies in the fact that the parameter $W$ is different for each sequence, which makes the problem non-trivial.
>
> We would like to stress that, to the best of our knowledge, the problem we study cannot be addressed using existing results. The wording "universal" is to be understood as universal within a class of functions, which here are autoregressive processes. We will change the title of the theorem, as we agree it is overclaiming the result. We apologize for this confusion.
>
> > Both the process of generating the augmented token $e_t^0$ from $x_{0:t}$, as well as the causal kernel descent iteration, theoretically involves an infinitely deep Transformer as $n \to \infty$. It might be more informative to discuss the theoretical implications of finite depth.
>
> Even though $e_t^0$ is obtained from $x_{0:t}$ by considering a limit, it is not an infinite depth limit but rather a scaling limit. The reviewer is absolutely correct that, to perfectly approximate the processes, we would need infinitely many layers. However, this is also true for standard gradient descent on any non-trivial optimization problem.
>
> We carefully discuss the theoretical implications of finite depth in Remark 1, as well as in l. 242 of Theorem 1.
>
> > The universality of next token predictability result is asymptotic in the context length $t$. Although the convergence is fast enough (exponential), most empirical observations are in the non-asymptotic regime in terms of context length. It might be good to explicitly provide analysis on this.
>
> We are happy to provide larger values of $t$ in the curve, even though we already consider $T = 15k$. Please note that it is computationally expensive to compute attention matrices for large sequences and that modern implementations use clever hardware-aware tricks that we did not use in our experiment. The goal of the experiment is to illustrate the theoretical results, which guarantee exponential convergence.
>
> **Questions**
>
> > Typo.
>
> Absolutely, thank you. Fixed!
>
> > In the experiment section...
>
> Yes, absolutely. Please note that $u_t^\star$ can be computed in closed form using Proposition 3, by solving a linear system.

---

> > ### Author Response · Authors · 2024-11-28
> >
> > Dear reviewer,
> >
> > The author-reviewer discussion period ends soon. We hope that our rebuttal has answered all your concerns. If not, please let us know, we would be happy to answer further questions.
> >
> > Thank you,
> >
> > The authors

---

> > > ### Comment · Reviewer_6GYT · 2024-11-28
> > >
> > > Dear authors,
> > >
> > > I have read the rebuttal and thank the authors for their detailed and candid responses.
> > >
> > > I maintain my positive opinion on this paper.

---

### Official Review · Reviewer_LWyw · 2024-11-04

**Soundness:** 3
**Presentation:** 3
**Contribution:** 3
**Rating:** 6
**Confidence:** 4

**Summary:**

The paper studies the approximation abilities of causal transformers when predicting the next token in-context for autoregressive sequences. In particular, it focuses on specific instances where the context-dependent map $f$ determining the next token $x_{t+1} = f(x_t)$ in the sequence belongs to the RKHS of a given kernel $k$ and is either linear or the sequence is periodic.  The authors introduce the causal kernel descent method and theoretically analyze its convergence properties. They provide a construction of the transformers layers with linear, exponential and softamx attention that can implement this method.

**Strengths:**

- The paper is well-written.
- The problem is conceptually well-motivated.
- The proposed method and analysis are clearly explained with all necessary details.
- The manuscript includes both rigorous proofs and empirical experiments that validate the findings.
- The proof technique and its connection to the Kaczmarz algorithm in Theorem 3 are insightful.

**Weaknesses:**

1. The choice of normalization in the attention appears to be specifically tailored to match the kernel used in the data-generating process (see Definition 2 and Theorem 1). This design decision raises questions about the generalizability of the findings, making it unclear how these results might extend to broader settings.
2. Since the analysis is confined to the class of functions determined by the chosen kernel, it is not clear how the results illustrated in the paper contribute towards understanding the universality of transformer models.
4. The theoretical results presented in the paper rely on taking the limits as the context length $t$ and the number of layers $n$ approach infinity. However,  the empirical experiments reported in Figure 4, demonstrate that Transformers with a finite number of layers, trained using the Adam optimizer, can achieve better performance than the proposed infinite-layer model. This discrepancy suggests that finite-sized trained Transformers may implement a different algorithm than the causal kernel descent method.
5. Exponential convergence for the case of exponential kernel and exponential or softamax attention requires periodic sequences.

**Questions:**

1. Could the authors clarify how their results using particular kernel choices and corresponding attention mechanisms contribute to understanding the universality of transformers in learning sequence-to-sequence functions and what they mean by universality in this context?
3. In Definition 2 could the authors clarify how is the periodicity enforced in instance (3).
4. In Proposition 1, it’s unclear what $n$ refers to, as elsewhere it denotes the number of layers, but here it relates only to the first layer that computes augmented tokens. In Appendix A.1, $n$ seems tied to the specific positional encoding. Could the authors clarify this?
5. Is it correct that in the presented model, the first layer computing the augmented tokens always uses softmax attention and does not have a skip connection, whereas the identical layers can employ different normalizations (softmax, linear, exponential) and include skip connections?
6.  In the paragraph: Augmented tokens,  could the authors clarify why $e_1^0 := (0_d,1,x_t,1,0_d,0_d)$ rather than $(0_d,1,0_d,1,x_t,0_d)$ based on the feed-forward map in appendix A.1.
7. (Minor) In Remark 1, there may be a typo in the definition of $\epsilon_1$.
8. In Equation 8 could the authors provide more insights regarding the interpretation of $\mu$ and its relation with the data-generating process?
9. For instance (4) described in section 5, did the authors try to train a model, and if so, what were the results?

---

> ### Author Response · Authors · 2024-11-23
>
> Thank you very much for your positive review and your suggestions. Please find our responses to your concerns below.
>
> **Weaknesses**
>
> > 1.
>
> It is true that while linear attention is particularly adapted to linear autoregressive processes, considering linear processes when using softmax attention is still restrictive. However, it remains a non-trivial problem because the matrices $\Omega$ in this case are different for each sequence. In our opinion, it is interesting to see that softmax attention can learn such processes.
>
> We would also like to emphasize that handling the row-wise normalization of the attention matrix is tricky, and we manage to address this in the article. An extension of our results would be to consider more general mappings $W \phi(x_t)$ directly.
>
> > 2.
>
> For the exponential kernel, the associated RKHS is universal, meaning that any function could be represented as $W \phi(x)$. However, we focus only on a subset of these functions that are linear in $x$, thereby proving universality within a smaller subset class of functions. This is reflected in the wording "towards understanding" in the title.
>
> We agree, however, that the title of our main Theorem 1 may be misleading, and we will revise it accordingly.
>
> > 3.
>
> Absolutely, the finite-depth model can outperform the infinitely deep model, even though the difference is small compared to the difference with the untrained finite-depth model. This is not surprising, as the trained model's parameters can be adjusted to model much faster optimization methods.
>
> We do not state in the paper that a trained Transformer necessarily implements the causal kernel descent. Instead, we propose this method as a tool to understand the expressivity of Transformers.
>
> **Questions**
>
> > 1.
>
> We prove that Transformers are universal within a certain class of functions. By universal, we mean that any function in this class can be approximated. Universality is always to be understood as universal within a specific function class.
>
> > 2.
>
> Periodicity is simply enforced by replicating sequences of a certain length.
>
> > 3.
>
> Thank you for spotting this inconsistency. In the context of the first layer, $n$ refers to a scaling factor. We will update the notation to avoid any confusion.
>
> > 4.
>
> Absolutely. Since skip connections are useful primarily when stacking many layers, we do not believe it is an issue to omit them in the first layer.
>
> > 5.
>
> Thank you for spotting this typo in the definition of $e^0_1$, which should indeed match your suggestion and is consistent with the expressions used in the proof. We will correct this typo on line 206.
>
> > 6.
>
> Yes, thank you. Fixed.
>
> > 7.
>
> Thank you for the interesting question. Actually, we provide an interpretation for $\mu$ using duality in Appendix B. We hope this clarifies the matter.
>
> > 8.
>
> No, we have not tried it. However, this is an excellent suggestion.

---

> > ### Author Response · Authors · 2024-11-28
> >
> > Dear reviewer,
> >
> > The author-reviewer discussion period ends soon. We hope that our rebuttal has answered all your concerns. If not, please let us know, we would be happy to answer further questions.
> >
> > Thank you,
> >
> > The authors

---

> > > ### Comment · Reviewer_LWyw · 2024-12-03
> > > **Further clarification**
> > >
> > > Thank you for your responses and clarifications to my previous questions; they have helped me better understand your work. I still have a question that I hope you can help clarify.
> > >
> > >
> > >  Given that in both instances (1 and 2), the family of functions is restricted to linear transformations $f(x) = W x $ and $f(x) = \Omega x $ with $ W, \Omega \in O(d) $, could the authors clarify whether there is a fundamental difference in the ground-truth autoregressive processes between these two instances? Is the main point of the analysis to demonstrate that the attention mechanism even with the softmax can learn (in-context) linear autoregressive processes effectively, and show how to handle the normalization, or is there a deeper distinctions between the two instances that impact their expressive power?

---

> > > > ### Author Response · Authors · 2024-12-04
> > > >
> > > > Dear reviewer,
> > > >
> > > > Thank you for your response.
> > > >
> > > > Yes, absolutely: when it comes to instance (2), our main contribution is to show that deep softmax-attention-only transformers can learn in context the linear autoregressive processes. In that sense, instances (1) and (2) have no fundamental differences in terms of ground truth autoregressive processes, but they significantly differ in terms of model architectures.
> > > >
> > > > Analyzing the linear autoregressive process with the non-linear attention kernel necessitates studying more complex operators, which we do by drawing connections to the convergence of the Kazmarz algorithm in infinite dimension. This corresponds to the proof of Theorem 3 in Appendix A.6. Note also that we manage to deal with the row-wise normalization induced by the softmax operation.
> > > >
> > > > That being said, we believe that the class of mappings $f$ for which softmax attention would succeed is much larger than linear ones: we prove it for periodic mappings (Theorem 4) and have numerical guarantees for instance (4) described in section 5.
> > > >
> > > > Thank you again for your review and constructive comments.
> > > >
> > > > Best,
> > > > The authors

---

### Official Review · Reviewer_Se7F · 2024-11-04

**Soundness:** 3
**Presentation:** 3
**Contribution:** 2
**Rating:** 3
**Confidence:** 4

**Summary:**

This paper considers the capacity of causal Transformers to the prediction of next token $x_{t+1}$ given autoregressive sequence $(x_1,\ldots,x_{t})$ as a prompt where $x_{t+1}=f(x_t)$. They explicitly construct a Transformer that learns the mapping $f$ in-context through a causal kernel descent method (which connects to the Kaczmarz algorithm) and prove long scope guarantees.

**Strengths:**

The strength comes mainly from the problem setup of studying next-token prediction, which seems to be quite an important task. The proof looks sound. The presentation of the work is clear. The authors perform empirical evaluation for their theoretical results. The discussion of theoretical results are provided in the work.

**Weaknesses:**

The major concern over this work is listed as follows.

(1) The problem setup is not realistic enough in the sequential relationship. In particular, the reviewer does not believe either the language or the time series admit simple relationship of $x_{t+1}=f(x_{t})$. Hence, the reviewer cannot understand how this work contributes to the understanding of Transformers on these modern ML tasks, which is believed to be very crucial to the ML community. Moreover, the reviewer believes that RNN might be able to do the same task as the results claimed in this work for Transformers. If this is true, the results might universally hold for many sequential models. Then, the unique advantage of Transformer is not highlighted.

(2) The problem setup is not realistic enough in the deterministic assumptions. The reviewer believes that either Markov chains or some random autoregressive models are necessary for the purpose of studying the universality of next token prediction of Transformers. At the current stage, this fully deterministic dynamic system seems rather naive.

(3) The limitations of this work is not properly discussed at the conclusion part.

**Questions:**

The questions and suggestions are given by the weakness section. The reviewer believes that this work requires significant improvement to reach his/her expectation and doesn't seem to be possibly addressed within the time frame of ICLR 2025 rebuttal phase.

However, if the author is able to resolve at least (1) or (2) in the question section, the reviewer might be able to improve his/her opinion.

---

> ### Author Response · Authors · 2024-11-23
>
> Thank you very much for reviewing our work and for the relevant remarks on the limitations of our approach. We will carefully highlight these limitations in the revised version of the submission.
>
> To specifically address your questions:
>
> **(1)**
>
> > The problem setup is not realistic enough in the sequential relationship. In particular, the reviewer does not believe either the language or the time series admit simple relationship of $x_{t+1} = f(x_t)$. Hence, the reviewer cannot understand how this work contributes to the understanding of Transformers on these modern ML tasks, which is believed to be very crucial to the ML community.
>
> We would like to emphasize that we specify in line 177 that higher-order recursions of the form $x_{t+1} = f(x_t, \cdots, x_{t-\tau})$ could be considered by embedding the tokens in a higher-dimensional space. More precisely, in this case, defining $y_t = (x_t, \cdots, x_{t-\tau})$, one has $y_{t+1}= (x_{t+1}, \cdots, x_{t+1-\tau}) = (f(x_t, \cdots, x_{t-\tau}), \cdots, x_{t+1-\tau}) $ which only depends on $y_t$ and can therefore be written as $F(y_t)$ for some function $F$. Therefore, by embedding the tokens in dimension $(\tau +1 )d$, our approach shows universality for autoregressive sequences of memory $\tau +1$. As such, as long as the recursion memory is finite, our approach can be generalized. We commit to add a precise remark about this fact in the paper. We hope we convince the reviewer on this point.
>
> The other crucial point we would like to emphasize is that, compared to previous theoretical works, we consider a strong dependency between the tokens in the sequence. This contrasts with all previous works on in-context learning, which assume i.i.d. data. Therefore, even though we acknowledge that our token encoding choice does not fully capture the structure of language, it is still better adapted than previous approaches.
>
> > Moreover, the reviewer believes that RNN might be able to do the same task as the results claimed in this work for Transformers. If this is true, the results might universally hold for many sequential models. Then, the unique advantage of Transformer is not highlighted.
>
> Even though we consider autoregressive sequences, it is not at all clear that RNNs can effectively capture these models. This is because estimating $W$ in context requires computations with long-range dependencies. We kindly remind the reviewer that the in-context function $f$ varies with each sequence and that determining the optimal $W$ requires inverting the data covariance matrix. This does not rigorously prove that RNNs cannot do it, but attention mechanisms inherently handle this, which is what we prove in our work.
>
> We believe RNNs would require more layers to "propagate" the information. While each RNN layer is less computationally expensive, the overall cost might be similar. Investigating this is complex and beyond the scope of this article. The key point is that we demonstrate how current Transformer-based architectures are particularly well-suited for in-context learning due to their global attention mechanism.
>
> **(2)**
>
> > The problem setup is not realistic enough in the deterministic assumptions. The reviewer believes that either Markov chains or some random autoregressive models are necessary for the purpose of studying the universality of next token prediction of Transformers. At the current stage, this fully deterministic dynamic system seems rather naive.
>
> We believe this is a very relevant critique. A possible extension could involve considering noisy dynamics of the form $x_{t+1} = f(x_t) + \varepsilon_t$. We will add a remark in this direction in the final version.
>
> We would like to stress that considering stochastic models would make the analysis even more challenging. Nevertheless, our results are not trivial and shed light on how attention mechanisms leverage pairwise interactions to estimate in-context quantities.
>
> We strongly hope that the reviewer will consider increasing their score as suggested in their review and would be happy to answer any further questions.

---

> > ### Author Response · Authors · 2024-11-24
> >
> > Dear reviewer,
> >
> > As said in our review, we modified the submission file to incorporate remarks related to point (1) (please see l. 288) as well as point (2), by adding a discussion paragraph in our conclusion (please see l. 533). We hope this can convince you to raise your score and recommend acceptance of our work. We are happy to provide any further clarifications. Thank you

---

> > > ### Author Response · Authors · 2024-11-26
> > >
> > > Dear reviewer,
> > >
> > > We have until tomorrow to update the pdf. Please let us know if the modifications we made satisfy you and if further clarifications are needed.
> > >
> > > Thanks a lot

---

> > > > ### Author Response · Authors · 2024-11-28
> > > >
> > > > Dear reviewer,
> > > >
> > > > The author-reviewer discussion period ends soon. Please let us know if you require further clarifications. We hope you can consider recommending acceptance of our work, as we provided answers for points (1) and (2) and modified the submission accordingly.
> > > >
> > > > Thanks a lot,
> > > >
> > > > The authors

---

### Official Review · Reviewer_gDtN · 2024-11-04

**Soundness:** 3
**Presentation:** 4
**Contribution:** 3
**Rating:** 6
**Confidence:** 3

**Summary:**

This paper extends a line of work showing what transformers are capable of learning. In particular, it shows that causal transformers are capable of learning kernelized linear functions, in context, which means that each token is the same linear function of the previous token. Previous work had focused on linear attention whereas this work extends to softmax attention. It also handles period sequences.

**Strengths:**

This paper furthers our understanding of why transformers, like the amazing LLMs, are able to learn in context. It extends to learning linear functions with softmax attention. It provides nontrivial, rigorous guarantees as well as experimental evidence corroborating the theory.

**Weaknesses:**

This is not a weakness of the work as much as a fit for the conference. This work seems important but also could arguably be more appropriate for a specialized theoretical audience like various theory conferences (COLT/ALT/STOC/FOCS). Previous works have covered linear functions and the extension to kernels and softmax attention is undoubtedly important for our understanding, but the question is how many of the conference attendees will appreciate these contributions. Of course, theory is a topic on the CFP and transformers are a key interest for ICLR so it's not a bad fit.

**Questions:**

Just to make sure I understand, is the main technical piece here handling the softmax activation? If the goal was to extend linear activation to the kernelized setting, would that just be adding the "kernel trick" to Sander et al (2024)?

---

> ### Author Response · Authors · 2024-11-23
>
> Thank you for your review and for taking the time to evaluate our work.
>
> > This is not a weakness of the work as much as a fit for the conference. This work seems important but also could arguably be more appropriate for a specialized theoretical audience like various theory conferences (COLT/ALT/STOC/FOCS). Previous works have covered linear functions and the extension to kernels and softmax attention is undoubtedly important for our understanding, but the question is how many of the conference attendees will appreciate these contributions. Of course, theory is a topic on the CFP and transformers are a key interest for ICLR so it's not a bad fit.
>
> We would like to emphasize that many theoretical works are published at ICLR or similar conferences such as NeurIPS and ICML, and we believe our work would be of interest to the audience. For instance, consider the following examples:
>
> - *A theoretical understanding of shallow vision Transformers*, ICLR 2024: [link](https://openreview.net/forum?id=jClGv3Qjhb)
>
> - *Generalization in diffusion models arises from geometry-adaptive harmonic representations*, a purely theoretical paper that received an outstanding award last year at ICLR 2024: [link](https://iclr.cc/virtual/2024/oral/19783)
>
> - *Emergence of clustering in attention models using PDE theory*, NeurIPS 2023: [link](https://openreview.net/forum?id=aMjaEkkXJx&referrer=%5Bthe%20profile%20of%20Yury%20Polyanskiy%5D(%2Fprofile%3Fid%3D~Yury_Polyanskiy1))
>
> - *Implicit bias in vision transformer models showing they can learn the spatial structure of images*, NeurIPS 2022: [link](https://proceedings.neurips.cc/paper_files/paper/2022/hash/f69707de866eb0805683d3521756b73f-Abstract-Conference.html)
>
> - *Attention layers provably solve single-location regression*, submitted this year to ICLR with good reviews: [link](https://openreview.net/forum?id=DVlPp7Jd7P)
>
> We would be very grateful if the reviewer could reconsider this remark in light of the examples provided. We are convinced that some quick investigation from the reviewer would confirm that our work fits the conference topics.
>
> > Just to make sure I understand, is the main technical piece here handling the softmax activation? If the goal was to extend linear activation to the kernelized setting, would that just be adding the "kernel trick" to Sander et al (2024)?
>
> We want to emphasize that our work differs significantly from Sander et al. (2024):
>
> - First, the reviewer is correct to observe that, in contrast to Sander et al. (2024), we consider softmax attention models. This makes the analysis not only more difficult but also much more aligned with practical applications.
> - Second, and importantly, we consider more general sequences with no commutativity assumptions on the context matrices $W$ or $\Omega$. This leads to a significantly more challenging problem.
> - Third, we consider deep non-linear Transformers, in contrast to Sander et al. (2024) who considered shallow linear Transformers. Having many layers is necessary in our case due to the non commutativity of the matrices $W$.
>
> Sander et al. (2024) focus on linear attention with one layer because the problem they study is simpler. Additionally, their work characterizes the minima of the loss landscape in that case, whereas we are interested in universality results. We stress that our work does not merely involve adding a "kernel trick"; we prove non-trivial results on the convergence of new causal kernel flows, which we believe can have a significant impact on the theoretical understanding of autoregressive attention-based models, such as LLMs.
>
> For these reasons, we would be grateful if the reviewer could consider increasing their score. We are happy to provide further clarifications should the reviewer have additional questions.

---

> > ### Comment · Reviewer_gDtN · 2024-11-23
> >
> > Thank you for the response. I'll raise my score slightly.

---

> > > ### Author Response · Authors · 2024-11-24
> > >
> > > Thank you very much for raising your score.

---

### Meta-Review · Area_Chair_uhxK · 2024-12-19

**Metareview:**

The paper studies the ability of causal transformers to perform next-token prediction in autoregressive sequences. The key contributions include a rigorous theoretical analysis showing how transformers implement causal kernel descent to learn linear and periodic functions asymptotically, as well as convergence guarantees in the infinite-layer and infinite-context regimes.

Strengths:

- The theoretical analysis is technically robust and advances our understanding of transformers' expressivity in next-token prediction.
- The proposed causal kernel descent method connects to well-established techniques like Kaczmarz iteration, offering new theoretical insights.
- Empirical results validate the theoretical findings, reinforcing the paper’s contributions.

Weaknesses:

- Claims of "universality" are overstated since the analysis focuses primarily on linear and periodic autoregressive processes.
- The theoretical setup (e.g., infinite layers and context) limits applicability to real-world transformers.
- Presentation is dense, with missing intuition for key constructions (e.g., augmented tokens) and unclear explanations of figures.

The paper was discussed extensively, with reviewers acknowledging both its strengths and limitations. While it is a clear borderline case, the consensus is that the technical rigor and theoretical insights justify weak acceptance, provided the claims are carefully framed in the final version.

**Additional Comments On Reviewer Discussion:**

N/A

---

### Decision · Program_Chairs · 2025-01-22

Accept (Poster)